# SELF-MONITORING NAVIGATION AGENT VIA AUXILIARY PROGRESS ESTIMATION

**Chih-Yao Ma**[*†], **Jiasen Lu**[*†], **Zuxuan Wu**[*‡], **Ghassan AlRegib**[†], **Zsolt Kira**[†],
**Richard Socher**[§] **& Caiming Xiong**[§]
[†]Georgia Institute of Technology
{cyma,jiasenlu,alregib,zkira}@gatech.edu
[‡]University of Maryland, College Park
{zxwu}@cs.umd.edu
[§]Salesforce Research
{rsocher,cxiong}@salesforce.com

## ABSTRACT

The Vision-and-Language Navigation (VLN) task entails an agent following navigational instruction in photo-realistic unknown environments. This challenging task demands that the agent be aware of which instruction was completed, which instruction is needed next, which way to go, and its navigation progress towards the goal. In this paper, we introduce a *self-monitoring* agent with two complementary components: (1) visual-textual co-grounding module to locate the instruction completed in the past, the instruction required for the next action, and the next moving direction from surrounding images and (2) progress monitor to ensure the grounded instruction correctly reflects the navigation progress. We test our self-monitoring agent on a standard benchmark and analyze our proposed approach through a series of ablation studies that elucidate the contributions of the primary components. Using our proposed method, we set the new state of the art by a significant margin (8% absolute increase in success rate on the unseen test set). Code is available at https://github.com/chihyaoma/selfmonitoring-agent.

## 1 INTRODUCTION

Recently, the Vision-and-Language (VLN) navigation task (Anderson et al., 2018b), which requires the agent to follow natural language instructions to navigate through a photo-realistic unknown environment, has received significant attention (Wang et al., 2018b; Fried et al., 2018). In the VLN task, an agent is placed in an unknown realistic environment and is required to follow natural language instructions to navigate from its starting location to a target location. In contrast to some existing navigation tasks (Kempka et al., 2016; Zhu et al., 2017; Mirowski et al., 2017; 2018), we address the class of tasks where the agent does not have an explicit representation of the target (e.g., location in a map or image representation of the goal) to know if the goal has been reached or not (Matuszek et al., 2013; Hemachandra et al., 2015; Duvallet et al., 2016; Arkin et al., 2017). Instead, the agent needs to be aware of its navigation status through the association between the sequence of observed visual inputs to instructions.

Consider an example as shown in Fig. 1, given the instruction "*Exit the bedroom and go towards the table. Go to the stairs on the left of the couch. Wait on the third step.*", the agent first needs to locate which instruction is needed for the next movement, which in turn requires the agent to be aware of (i.e., to explicitly represent or have an attentional focus on) which instructions were completed or ongoing in the previous steps. For instance, the action *"Go to the stairs"* should be carried out once the agent has exited the room and moved towards the table. However, there exists inherent ambiguity for "*go towards the table*". Intuitively, the agent is expected to *"Go to the stairs"* after completing "*go towards the table*". But, it is not clear what defines the completeness of *"Go towards the table"*. The completeness of an ongoing action often depends on the availability of the next action. Since the transition between past and next part of the instructions is a *soft*

---

[*]Work done while the authors were research interns at Salesforce Research.

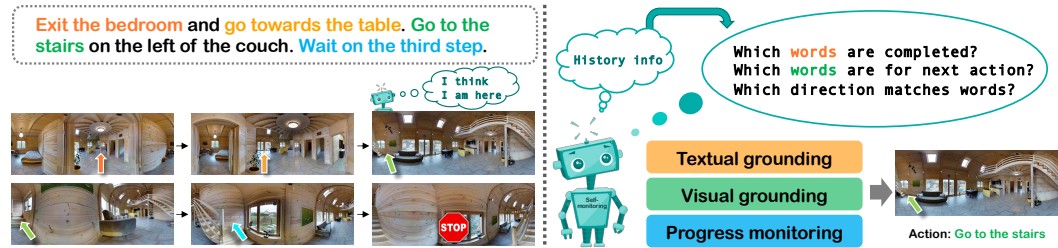

Figure 1: Vision-and-Language Navigation task and our proposed self-monitoring agent. The agent is constantly aware of what was completed, what is next, and where to go, as it navigates through unknown environments by following navigational instructions.

*boundary*, in order to determine when to transit and to follow the instruction correctly the agent is required to keep track of both grounded instructions. On the other hand, assessing the progress made towards the goal has indeed been shown to be important for goal-directed tasks in humans decision-making (Benn et al., 2014; Chatham et al., 2012; Berkman & Lieberman, 2009). While a number of approaches have been proposed for VLN (Anderson et al., 2018b; Wang et al., 2018b; Fried et al., 2018), previous approaches generally are not aware of which instruction is next nor progress towards the goal; indeed, we qualitatively show that even the attentional mechanism of the baseline does not successfully track this information through time.

In this paper, we propose an agent endowed with the following abilities: (1) identify which direction to go by finding the part of the instruction that corresponds to the observed images—*visual grounding*, (2) identify which part of the instruction has been completed or ongoing and which part is potentially needed for the next action selection—*textual grounding*, and (3) ensure that the grounded instruction can correctly be used to estimate the progress made towards the goal, and apply regularization to ensure this —*progress monitoring*. Therefore, we introduce the self-monitoring agent consisting of two complementary modules: visual-textual co-grounding and progress monitor.

More specifically, we achieve both visual and textual grounding simultaneously by incorporating the full history of grounded instruction, observed images, and selected actions into the agent. We leverage the structural bias between the words in instructions used for action selection and progress made towards the goal and propose a new objective function for the agent to measure how well it can estimate the completeness of instruction-following. We then demonstrate that by conditioning on the positions and weights of grounded instruction as input, the agent can be self-monitoring of its progress and further ensure that the textual grounding accurately reflects the progress made.

Overall, we propose a novel *self-monitoring* agent for VLN and make the following contributions: (1) We introduce the **visual-textual co-grounding** module, which performs grounding interdependently across both visual and textual modalities. We show that it can outperform the baseline method by a large margin. (2) We propose to equip the self-monitoring agent with a **progress monitor**, and for navigation tasks involving instructions instantiate this by introducing a new objective function for training. We demonstrate that, unlike the baseline method, the position of grounded instruction can follow both past and future instructions, thereby tracking progress to the goal. (3) With the proposed self-monitoring agent, we set the new state-of-the-art performance on both seen and unseen environments on the standard benchmark. With 8% absolute improvement in success rate on the unseen test set, we are ranked #1 on the challenge leaderboard.

## 2 SELF-MONITORING NAVIGATION AGENT

### 2.1 NOTATION

Given a natural language instruction with $L$ words, its representation is denoted by $X = \{x_1, x_2, \ldots, x_L\}$, where $x_l$ is the feature vector for the $l$-th word encoded by an LSTM language encoder. Following Fried et al. (2018), we enable the agent with panoramic view. At each time step, the agent perceives a set of images at each viewpoint $v_t = \{v_{t,1}, v_{t,2}, ..., v_{t,K}\}$, where $K$

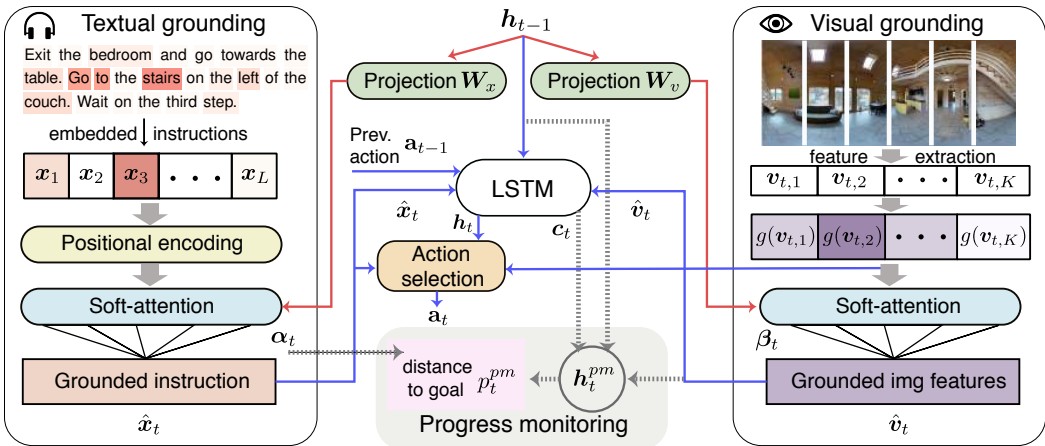

Figure 2: Proposed self-monitoring agent consisting of visual-textual co-grounding, progress monitoring, and action selection modules. *Textual grounding*: identify which part of the instruction has been completed or ongoing and which part is potentially needed for next action. *Visual grounding*: summarize the observed surrounding images. *Progress monitor*: regularize and ensure grounded instruction reflects progress towards the goal. *Action selection*: identify which direction to go.

is the maximum number of navigable directions[1], and $\boldsymbol{v}_{t,k}$ represents the image feature of direction $k$. The co-grounding feature of instruction and image are denoted as $\hat{\boldsymbol{x}}_t$ and $\hat{\boldsymbol{v}}_t$ respectively. The selected action is denoted as $\mathbf{a}_t$. The learnable weights are denoted with $\boldsymbol{W}$, with appropriate sub/super-scripts as necessary. We omit the bias term $\boldsymbol{b}$ to avoid notational clutter in the exposition.

## 2.2 VISUAL AND TEXTUAL CO-GROUNDING

First, we propose a visual and textual co-grounding model for the vision and language navigation task, as illustrated in Fig. 2. We model the agent with a sequence-to-sequence architecture with attention by using a recurrent neural network. More specifically, we use Long Short Term Memory (LSTM) to carry the flow of information effectively. At each step $t$, the decoder observes representations of the current attended panoramic image feature $\hat{\boldsymbol{v}}_t$, previous selected action $\mathbf{a}_{t-1}$ and current grounded instruction feature $\hat{\boldsymbol{x}}_t$ as input, and outputs an encoder context $\boldsymbol{h}_t$:

$$\boldsymbol{h}_t = LSTM([\hat{\boldsymbol{x}}_t, \hat{\boldsymbol{v}}_t, \mathbf{a}_{t-1}]) \tag{1}$$

where $[,]$ denotes concatenation. The previous encoder context $\boldsymbol{h}_{t-1}$ is used to obtain the textual grounding feature $\hat{\boldsymbol{x}}_t$ and visual grounding feature $\hat{\boldsymbol{v}}_t$, whereas we use current encoder context $\boldsymbol{h}_t$ to obtain next action $\mathbf{a}_t$, all of which will be illustrated in the rest of the section.

**Textual grounding.** When the agent moves from one viewpoint to another, it is required to identify which direction to go by relying on a grounded instruction, i.e. which parts of the instruction should be used. This can either be the instruction matched with the past (ongoing action) or predicted for the future (next action). To capture the relative position between words within an instruction, we incorporate the positional encoding $PE(\cdot)$ (Vaswani et al., 2017) into the instruction features. We then perform soft-attention on the instruction features $\boldsymbol{X}$, as shown on the left side of Fig. 2. The attention distribution over $L$ words of the instructions is computed as:

$$z_{t,l}^{\text{textual}} = (\boldsymbol{W}_x \boldsymbol{h}_{t-1})^\top PE(\boldsymbol{x}_l), \quad \text{and} \quad \boldsymbol{\alpha}_t = \text{softmax}(\boldsymbol{z}_t^{\text{textual}}), \tag{2}$$

where $\boldsymbol{W}_x$ are parameters to be learnt. $z_{t,l}^{\text{textual}}$ is a scalar value computed as the correlation between word $l$ of the instruction and previous hidden state $\boldsymbol{h}_{t-1}$, and $\boldsymbol{\alpha}_t$ is the attention weight over features in $\boldsymbol{X}$ at time $t$. Based on the textual attention distribution, the grounded textual feature $\hat{\boldsymbol{x}}_t$ can be obtained by the weighted sum over the textual features $\hat{\boldsymbol{x}}_t = \boldsymbol{\alpha}_t^T \boldsymbol{X}$.

---

[1]Empirically, we found that using only the images on navigable directions to be slightly better than using all 36 surrounding images (12 headings $\times$ 3 elevations with 30 degree intervals).

**Visual grounding.** In order to locate the completed or ongoing instruction, the agent needs to keep track of the sequence of images observed along the navigation trajectory. We thus perform visual attention over the surrounding views based on its previous hidden vector $\boldsymbol{h}_{t-1}$. The visual attention weight $\boldsymbol{\beta}_t$ can be obtained as:

$$z_{t,k}^{\text{visual}} = (\boldsymbol{W}_v \boldsymbol{h}_{t-1})^\top g(\boldsymbol{v}_{t,k}), \quad \text{and} \quad \boldsymbol{\beta}_t = \text{softmax}(\boldsymbol{z}_t^{\text{visual}}), \tag{3}$$

where $g$ is a one-layer Multi-Layer Perceptron (MLP), $\boldsymbol{W}_v$ are parameters to be learnt. Similar to Eq. 2, the grounded visual feature $\hat{\boldsymbol{v}}_t$ can be obtained by the weighted sum over the visual features $\hat{\boldsymbol{v}}_t = \boldsymbol{\beta}_t^T \boldsymbol{V}$.

**Action selection.** To make a decision on which direction to go, the agent finds the image features on navigable directions with the highest correlation with the grounded navigation instruction $\hat{\boldsymbol{x}}_t$ and the current hidden state $\boldsymbol{h}_t$. We use the inner-product to compute the correlation, and the probability of each navigable direction is then computed as:

$$o_{t,k} = (\boldsymbol{W}_a[\boldsymbol{h}_t, \hat{\boldsymbol{x}}_t])^\top g(\boldsymbol{v}_{t,k}) \quad \text{and} \quad \boldsymbol{p}_t = \text{softmax}(\boldsymbol{o}_t), \tag{4}$$

where $\boldsymbol{W}_a$ are the learnt parameters, $g(\cdot)$ is the same MLP as in Eq. 3, and $\boldsymbol{p}_t$ is the probability of each navigable direction at time $t$. We use categorical sampling during training to select the next action $\mathbf{a}_t$. Unlike the previous method with the panoramic view (Fried et al., 2018), which attends to instructions only based on the history of observed images, we achieve both textual and visual grounding using the shared hidden state output containing grounded information from both textual and visual modalities. During action selection, we rely on both hidden state output and grounded instruction, instead of only relying on grounded instruction.

## 2.3 PROGRESS MONITOR

It is imperative that the textual-grounding correctly reflects the progress towards the goal, since the agent can then implicitly know where it is now and what the next instruction to be completed will be. In the **visual-textual co-grounding** module, we ensure that the grounded instruction reasonably informs decision making when selecting a navigable direction. This is necessary but not sufficient for ensuring that the notion of progress to the goal is encoded. Thus, we propose to equip the agent with a **progress monitor** that serves as regularizer during training and prunes unfinished trajectories during inference.

Since the positions of localized instruction can be a strong indication of the navigation progress due to the structural alignment bias between navigation steps and instruction, the progress monitor can estimate how close the current viewpoint is to the final goal by conditioning on the positions and weights of grounded instruction. This can further enforce the result of textual-grounding to align with the progress made towards the goal and to ensure the correctness of the textual-grounding.

The progress monitor aims to estimate the navigation progress by conditioning on three inputs: the history of grounded images and instructions, the current observation of the surrounding images, and the positions of grounded instructions. We therefore represent these inputs by using (1) the previous hidden state $\boldsymbol{h}_{t-1}$ and the current cell state $\boldsymbol{c}_t$ of the LSTM, (2) the grounded surrounding images $\hat{\boldsymbol{v}}_t$, and (3) the distribution of attention weights of textual-grounding $\boldsymbol{\alpha}_t$, as shown at the bottom of Fig. 2 represented by dotted lines.

Our proposed progress monitor first computes an additional hidden state output $\boldsymbol{h}_t^{pm}$ by using grounded image representations $\hat{\boldsymbol{v}}_t$ as input, similar to how a regular LSTM computes hidden states except we use concatenation over element-wise addition for empirical reasons[2]. The hidden state output is then concatenated with the attention weights $\boldsymbol{\alpha}_t$ on textual-grounding to estimate how close the agent is to the goal[3]. The output of the progress monitor $p_t^{pm}$, which represents the completeness of instruction-following, is computed as:

$$\boldsymbol{h}_t^{pm} = \sigma(\boldsymbol{W}_h([\boldsymbol{h}_{t-1}, \hat{\boldsymbol{v}}_t]) \otimes tanh(\boldsymbol{c}_t)), \quad p_t^{pm} = tanh(\boldsymbol{W}_{pm}([\boldsymbol{\alpha}_t, \boldsymbol{h}_t^{pm}])) \tag{5}$$

where $\boldsymbol{W}_h$ and $\boldsymbol{W}_{pm}$ are the learnt parameters, $c_t$ is the cell state of the LSTM, $\otimes$ denotes the element-wise product, and $\sigma$ is the sigmoid function.

---

[2]We found that using concatenation provides slightly better performance and stable training.
[3]We use zero-padding to handle instructions with various lengths.

## 2.4 TRAINING AND INFERENCE

**Training.** We introduce a new objective function to train the proposed progress monitor. The training target $y_t^{pm}$ is defined as the normalized distance in units of length from the current viewpoint to the goal, i.e., the target will be $0$ at the beginning and closer to $1$ as the agent approaches the goal[4]. Note that the target can also be lower than $0$, if the agent's current distance from the goal is farther than the starting point. Finally, our self-monitoring agent is optimized with a cross-entropy loss and a mean squared error loss, computed with respect to the outputs from both action selection and progress monitor.

$$\mathcal{L}_{loss} = -\lambda \underbrace{\sum_{t=1}^{T} y_t^{nv} log(p_{k,t})}_{\text{action selection}} - (1-\lambda) \underbrace{\sum_{t=1}^{T} (y_t^{pm} - p_t^{pm})^2}_{\text{progress monitor}} \qquad (6)$$

where $p_{k,t}$ is the action probability of each navigable direction, $\lambda = 0.5$ is the weight balancing the two losses, and $y_t^{nv}$ is the ground-truth navigable direction at step $t$.

**Inference.** During inference, we follow Fried et al. (2018) by using beam search. we propose that, while the agent decides which trajectories in the beams to keep, it is equally important to evaluate the state of the beams on actions as well as on the agent's confidence in completing the given instruction at each traversed viewpoint. We accomplish this idea by integrating the output of our progress monitor into the accumulated probability of beam search. At each step, when candidate trajectories compete based on accumulated probability, we integrate the estimated completeness of instruction-following $p_t^{pm}$ (normalized between 0 to 1) with action probability $p_{k,t}$ to directly evaluate the partial and unfinished candidate routes: $p_t^{beam} = p_t^{pm} \times p_{k,t}$.

Without beam search, we use greedy decoding for action selection with one condition. If the progress monitor output decreases ($p_{t+1}^{pm} < p_t^{pm}$), the agent is required to move back to the previous viewpoint and select the action with next highest probability. We repeat this process until the selected action leads to increasing progress monitor output. We denote this procedure as *progress inference*.

## 3 EXPERIMENTS

**R2R Dataset.** We use the Room-to-Room (R2R) dataset (Anderson et al., 2018b) for evaluating our proposed approach. The R2R dataset is built upon the Matterport3D dataset (Chang et al., 2017) and has 7,189 paths sampled from its navigation graphs. Each path has three ground-truth navigation instructions written by humans. The whole dataset is divided into 4 sets: training, validation seen, validation unseen, and test sets unseen.

**Evaluation metrics.** We follow the same evaluation metrics used by previous work on the R2R task: (1) Navigation Error (NE), mean of the shortest path distance in meters between the agent's final

---

[4]We set the target to 1 if the agent's distance to the goal is less than 3.

Table 1: Performance comparison with the state of arts: Student-forcing (Anderson et al., 2018b), RPA (Wang et al., 2018b), and Speaker-Follower (Fried et al., 2018). *: with data augmentation. leaderboard: when using beam search, we modify our search procedure to comply with the leaderboard guidelines, i.e., all traversed viewpoints are recorded.

| Method | Validation-Seen | | | | Validation-Unseen | | | | Test (unseen) | | | |
|---|---|---|---|---|---|---|---|---|---|---|---|---|
| | NE ↓ | SR ↑ | OSR ↑ | SPL ↑ | NE ↓ | SR ↑ | OSR ↑ | SPL ↑ | NE ↓ | SR ↑ | OSR ↑ | SPL ↑ |
| Random | 9.45 | 0.16 | 0.21 | - | 9.23 | 0.16 | 0.22 | - | 9.77 | 0.13 | 0.18 | - |
| Student-forcing | 6.01 | 0.39 | 0.53 | - | 7.81 | 0.22 | 0.28 | - | 7.85 | 0.20 | 0.27 | - |
| RPA | 5.56 | 0.43 | 0.53 | - | 7.65 | 0.25 | 0.32 | - | 7.53 | 0.25 | 0.33 | - |
| Speaker-Follower | 3.88 | 0.63 | 0.71 | - | 5.24 | 0.50 | 0.63 | - | - | - | - | - |
| Speaker-Follower* | 3.08 | 0.70 | **0.78** | - | 4.83 | 0.55 | 0.65 | - | 4.87 | 0.53 | 0.64 | - |
| (leaderboard) | - | - | - | - | - | - | - | - | 4.87 | 0.53 | 0.96 | 0.01 |
| Ours (beam search) | 3.23 | 0.70 | **0.78** | 0.66 | 5.04 | 0.57 | **0.70** | 0.51 | 4.99 | 0.57 | 0.68 | 0.51 |
| (leaderboard) | - | - | - | - | - | - | - | - | 4.99 | 0.57 | 0.95 | 0.02 |
| Ours* (beam search) | **3.04** | **0.71** | 0.78 | 0.67 | **4.62** | **0.58** | 0.68 | **0.52** | **4.48** | **0.61** | 0.70 | **0.56** |
| (leaderboard) | - | - | - | - | - | - | - | - | **4.48** | **0.61** | **0.97** | 0.02 |

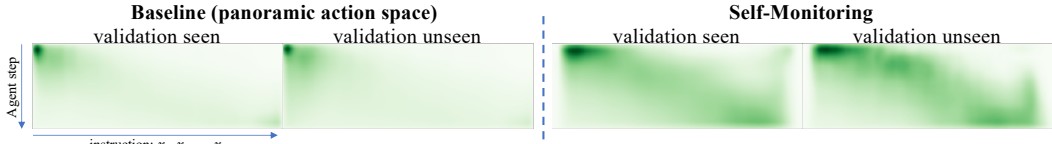

Figure 3: The positions and weights of grounded instructions as agents navigate by following instructions. Our self-monitoring agent with progress monitor demonstrates the grounded instruction used for action selection shifts gradually from the beginning of instructions towards the end. This is not true of the baseline method.

position and the goal location. (2) Success Rate (SR), the percentage of final positions less than 3m away from the goal location. (3) Oracle Success Rate (OSR), the success rate if the agent can stop at the closest point to the goal along its trajectory. In addition, we also include the recently introduced Success rate weighted by (normalized inverse) Path Length (SPL) (Anderson et al., 2018a), which trades-off Success Rate against trajectory length.

## 3.1 COMPARISON WITH PRIOR ART

We first compare the proposed self-monitoring agent with existing approaches. As shown in Table 1, our method achieves significant performance improvement compared to the state of the arts without data augmentation. We achieve 70% SR on the seen environment and 57% on the unseen environment while the existing best performing method achieved 63% and 50% SR respectively. When trained with synthetic data[5], our approach achieves slightly better performance on the seen environments and significantly better performance on both the validation unseen environments and the test unseen environments when submitted to the test server. We achieve 3% and 8% improvement on SR on both validation and test unseen environments. Both results with or without data augmentation indicate that our proposed approach is more generalizable to unseen environments. At the time of writing, our self-monitoring agent is ranked #1 on the challenge leader-board among the state of the arts.

Note that both Speaker-Follower and our approach in Table 1 use beam search. For comparison without using beam search, please refer to the Appendix.

**Textually grounded agent.** Intuitively, an instruction-following agent is required to strongly demonstrate the ability to correctly focus and follow the corresponding part of the instruction as it navigates through an environment. We thus record the distribution of attention weights on instruction at each step as indications of which parts of the instruction being used for action selection. We average all runs across both validation seen and unseen dataset splits. Ideally, we expect to see the distribution of attention weights lies close to a diagonal, where at the beginning, the agent focuses on the beginning of the instruction and shifts its attention towards the end of instruction as it moves closer to the goal.

To demonstrate, we use the method with panoramic action space proposed in Fried et al. (2018) as a baseline for comparison. As shown in Figure 3, our self-monitoring agent with progress monitor demonstrates that the positions of grounded instruction over time form a line similar to a diagonal. This result may further indicate that the agent successfully utilizes the attention on instruction to complete the task sequentially. We can also see that both agents were able to focus on the first part of the instruction at the beginning of navigation consistently. However, as the agent moves further in unknown environments, our self-monitoring agent can still successfully identify the parts of instruction that are potentially useful for action selection, whereas the baseline approach becomes uncertain about which part of the instruction should be used for selecting an action.

---

[5]We use the exact same synthetic data generated from the Speaker as in Fried et al. (2018) for comparison.

Table 2: Ablation study showing the effect of each proposed component. All methods use the panoramic action space. Note that, for methods using beam search during inference, only the last selected trajectory is used for evaluating OSR and SPL. *: we implemented the model from Speaker-Follower (Fried et al., 2018) with panoramic action space as baseline.

| # | Co-Grounding | Progress Monitor | Greedy Decoding | Progress Inference | Beam Search | Data Aug. | NE ↓ | SR ↑ | OSR ↑ | SPL ↑ | NE ↓ | SR ↑ | OSR ↑ | SPL ↑ |
|---|---|---|---|---|---|---|---|---|---|---|---|---|---|---|
| | | | | | | | | Validation-Seen | | | | Validation-Unseen | | |
| Baseline* | | | | | | | 4.36 | 0.54 | 0.68 | - | 7.22 | 0.27 | 0.39 | - |
| 1 | ✓ | | ✓ | | | | 3.65 | 0.65 | 0.75 | 0.56 | 6.07 | 0.42 | 0.57 | 0.28 |
| 2 | ✓ | ✓ | ✓ | | | | 3.72 | 0.63 | 0.75 | 0.56 | 5.98 | 0.44 | 0.58 | 0.30 |
| 3 | ✓ | ✓ | ✓ | | | ✓ | 3.22 | 0.67 | **0.78** | 0.58 | 5.52 | 0.45 | 0.56 | 0.32 |
| 4 | ✓ | ✓ | | ✓ | | | 3.56 | 0.65 | 0.75 | 0.58 | 5.89 | 0.46 | 0.60 | 0.32 |
| 5 | ✓ | ✓ | | ✓ | | ✓ | 3.18 | 0.68 | 0.77 | 0.58 | 5.41 | 0.47 | 0.59 | 0.34 |
| 6 | ✓ | | | | ✓ | | 3.66 | 0.66 | 0.76 | 0.62 | 5.70 | 0.49 | 0.68 | 0.42 |
| 7 | ✓ | ✓ | | | ✓ | | 3.23 | 0.70 | **0.78** | 0.66 | 5.04 | 0.57 | **0.70** | 0.51 |
| 8 | ✓ | ✓ | | | ✓ | ✓ | **3.04** | **0.71** | 0.78 | **0.67** | **4.62** | **0.58** | 0.68 | **0.52** |

## 3.2 ABLATION STUDY

We now discuss the importance of each component proposed in this work. We begin with the same baseline as before (agent with panoramic action space in Fried et al. (2018))[6].

**Co-grounding.** When comparing the baseline with row #1 in our proposed method, we can see that our co-grounding agent outperformed the baseline with a large margin. This is due to the fact that we use the LSTM to carry both the textually and visually grounded content, and the decision on each navigable direction is predicted with both textually grounded instruction and the hidden state output of the LSTM. On the other hand, the baseline agent relies on the LSTM to carry visually grounded content, and uses the hidden state output for predicting the textually grounded instruction. As a result, we observed that instead of predicting the instruction needed for selecting a navigable direction, the textually grounded instruction may match with the past sequence of observed images implicitly saved within the LSTM.

**Progress monitor.** Given the effective co-grounding, the proposed progress monitor further ensure that the grounded instruction correctly reflects the progress made toward the goal. This further improves the performance especially on the unseen environments as we can see from row #1 and #2.

When using the progress inference, the progress monitor serve as a progress indicator for the agent to decide when to move back to the last viewpoint. We can see from row #2 and #4 that the SR performance can be further improved around 2% on both seen and unseen environments.

Finally, we integrate the output of the progress monitor with the state-factored beam search (Fried et al., 2018), so that the candidate paths compete not only based on the probability of selecting a certain navigable direction but also on the estimated correspondence between the past trajectory and the instruction. As we can see by comparing row #2, #6, and #7, the progress monitor significantly improved the success rate on both seen and unseen environments and is the key for surpassing the state of the arts even without data augmentation. We can also see that when using beam search without progress monitor, the SR on unseen improved 7% (row #1 vs #6), while using beam search integrated with progress estimation improved 13% (row #2 vs #7).

**Data augmentation.** In the above, we have shown each row in our approach contributes to the performance. Each of them increases the success rate and reduces the navigation error incrementally. By further combining them with the data augmentation pre-trained from the *speaker* (Fried et al., 2018), the SR and OSR are further increased, and the NE is also drastically reduced. Interestingly, the performance improvement introduced by data augmentation is smaller than from Speaker-Follower on the validation sets (see Table 1 for comparison). This demonstrates that our proposed method is more data-efficient.

---

[6]Note that our results for this baseline are slightly higher on val-seen and slightly lower on val-unseen than those reported, due to differences in hyper-parameter choices.

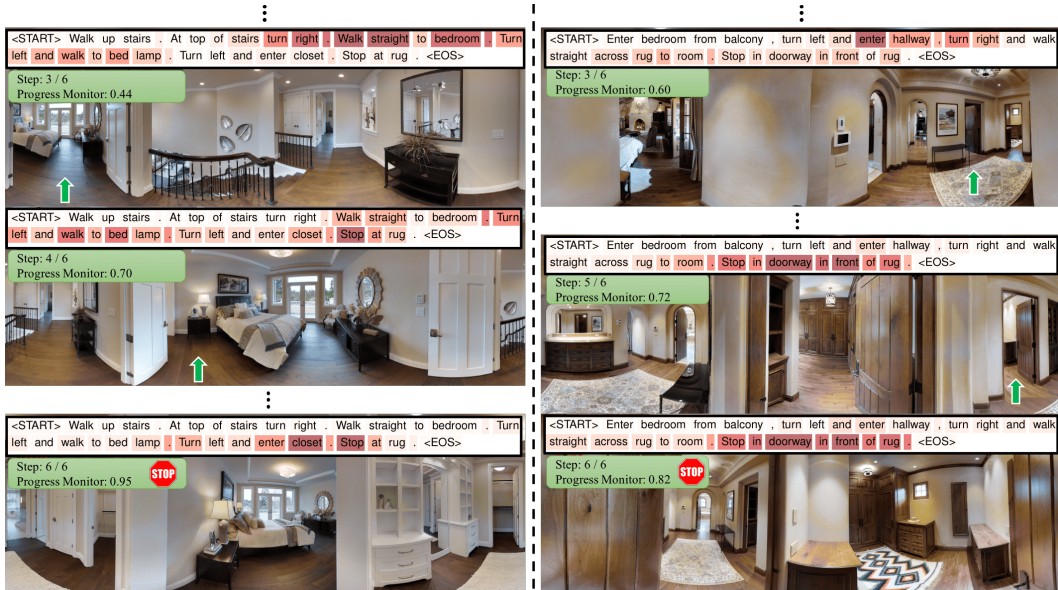

Figure 4: Successful self-monitoring agent navigates in two unseen environments. The agent is able to correctly follow the grounded instruction and achieve the goal successfully. The percentage of instruction completeness estimated by the proposed progress monitor gradually increases as the agent navigates and approaches the goal. Finally, the agent grounded the word "*Stop*" to stop (see the supplementary material for full figures).

### 3.3 QUALITATIVE RESULTS

To further validate the proposed method, we qualitatively show how the agent navigates through unseen environments by following instructions as shown in Fig. 4. In each figure, the agent follows the grounded instruction (at the top of the figure) and decides to move towards a certain direction (green arrow). For the full figures and more examples of successful and failed agents in both unseen and seen environments, please see the supplementary material.

Consider the trajectory on the left side in Fig. 4, at step 3, the grounded instruction illustrated that the agent just completed "*turn right*" and focuses mainly on "*walk straight to bedroom*". As the agent entered the bedroom, it then shifts the textual grounding to the next action "*Turn left and walk to bed lamp*". Finally, at step 6, the agent completed another "*turn left*" and successfully stop at the rug (see the supplementary material for the importance of dealing with duplicate actions). Consider the example on the right side, the agent has already entered the hallway and now turns right to walk across to another room. However, it is ambiguous that which room the instructor is referring to. At step 5, our agent checked out the room on the left first and realized that it does not match with "*Stop in doorway in front of rug*". It then moves to the next room and successfully stops at the goal.

In both cases, we can see that the completeness estimated by progress monitor gradually increases as the agent steadily navigates toward the goal. We have also observed that the estimated completeness ends up much lower for failure cases (see the supplementary material for further details).

## 4 RELATED WORK

**Vision, Language, and Navigation.**. There is a plethora work investigating the combination of vision and language for a multitude of applications (Zhou et al., 2018a;b; Antol et al., 2015; Tapaswi et al., 2016; Das et al., 2017), etc. While success has been achieved in these tasks to handle massive corpora of *static* visual input and text data, a resurgence of interest focuses on equipping an agent with the ability to interact with its surrounding environment for a particular goal such as object manipulation with instructions (Misra et al., 2016; Arkin et al., 2017), grounded language acquisition (Al-Omari et al., 2017; Kollar et al., 2013; Spranger & Steels, 2015; Dubba et al., 2014),

embodied question answering (Das et al., 2018; Gordon et al., 2018), and navigation (Matuszek et al., 2013; Hemachandra et al., 2015; Duvallet et al., 2016; Zhu et al., 2017; de Vries et al., 2018; Yuke Zhu, 2017; Mousavian et al., 2018; Wayne et al., 2018; Wang et al., 2018a; Mirowski et al., 2017; 2018; Zamir et al., 2018). In this work, we concentrate on the recently proposed the Vision-and-Language Navigation task (Anderson et al., 2018b)—asking an agent to carry out sophisticated natural-language instructions in a 3D environment. This task has application to fields such as robotics; in contrast to traditional map-based navigation systems, navigation with instructions provides a flexible way to generalize across different environments.

A few approaches have been proposed for the VLN task. For example, Anderson et al. (2018b) address the task in the form of a sequence-to-sequence translation model. Yu et al. (2018) introduce a guided feature transformation for textual grounding. Wang et al. (2018b) present a planned-head module by combing model-free and model-based reinforcement learning approaches. Recently, Fried et al. (2018) propose to train a *speaker* to synthesize new instructions for data augmentation and further use it for pragmatic inference to rank the candidate routes. These approaches leverage attentional mechanisms to select related words from a given instruction when choosing an action, but those agents are deployed to explore the environment without knowing about what progress has been made and how far away the goal is. In this paper, we propose a self-monitoring agent that performs co-grounding on both visual and textual inputs and constantly monitors its own progress toward the goal as a way of regularizing the textual grounding.

**Visual and textual grounding.** Visual grounding learns to localize the most relevant object or region in an image given linguistic descriptions, and has been demonstrated as an essential component for a variety of vision tasks like image captioning (Hu et al., 2016; Rohrbach et al., 2016; Lu et al., 2018), visual question answering (Lu et al., 2016b; Agrawal et al., 2018), relationship detection (Lu et al., 2016a; Ma et al., 2018) and referral expression (Nagaraja et al., 2016; Gavrilyuk et al., 2018). In contrast to identifying regions or objects, we perform visual grounding to locate relevant images (views) in a panoramic photo constructed by stitching multiple images with the aim of choosing which direction to go. Extensive efforts have been made to ground language instructions into a sequence of actions (MacMahon et al., 2006; Branavan et al., 2009; Vogel & Jurafsky, 2010; Tellex et al., 2011; Artzi & Zettlemoyer, 2013; Andreas & Klein, 2015; Mei et al., 2016; Cohn et al., 2016; Misra et al., 2017). These early approaches mainly emphasize the incorporation of structural alignment biases between the linguistic structure and sequence of actions (Mei et al., 2016; Andreas & Klein, 2015), and assume the agents are in relatively easy environment where limited visual perception is required to fulfill the instructions.

## 5 CONCLUSION

We introduce a *self-monitoring* agent which consists of two complementary modules: visual-textual co-grounding module and progress monitor. The visual-textual co-grounding module locates the instruction completed in the past, the instruction needed in the next action, and the moving direction from surrounding images. The progress monitor regularizes and ensures the grounded instruction correctly reflects the progress towards the goal by explicitly estimating the completeness of instruction-following. This estimation is conditioned on the positions and weights of grounded instruction. Our approach sets a new state-of-the-art performance on the standard Room-to-Room dataset on both seen and unseen environments. While we present one instantiation of self-monitoring for a decision-making agent, we believe that this concept can be applied to other domains as well.

## ACKNOWLEDGMENTS

This research was partially supported by DARPAs Lifelong Learning Machines (L2M) program, under Cooperative Agreement HR0011-18-2-001. We thank the authors from Fried et al. (2018), Ronghang Hu and Daniel Fried, for communicating with us and providing details of the implementation and synthetic instructions for fair comparison.

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

Table 3: Performance comparison with the state of arts *without beam search*: Student-forcing (Anderson et al., 2018b), RPA (Wang et al., 2018b), and Speaker-Follower (Fried et al., 2018). *: with data augmentation.

| Method | Validation-Seen | | | | Validation-Unseen | | | | Test (unseen) | | | |
|---|---|---|---|---|---|---|---|---|---|---|---|---|
| | NE ↓ | SR ↑ | OSR ↑ | SPL ↑ | NE ↓ | SR ↑ | OSR ↑ | SPL ↑ | NE ↓ | SR ↑ | OSR ↑ | SPL ↑ |
| Random | 9.45 | 0.16 | 0.21 | - | 9.23 | 0.16 | 0.22 | - | 9.77 | 0.13 | 0.18 | - |
| Student-forcing | 6.01 | 0.39 | 0.53 | - | 7.81 | 0.22 | 0.28 | - | 7.85 | 0.20 | 0.27 | - |
| RPA | 5.56 | 0.43 | 0.53 | - | 7.65 | 0.25 | 0.32 | - | 7.53 | 0.25 | 0.33 | - |
| Speaker-Follower* | 3.36 | 0.66 | 0.74 | - | 6.62 | 0.36 | 0.45 | - | 6.62 | 0.35 | 0.44 | 0.28 |
| Ours* (Greedy Decoding) | 3.22 | 0.67 | **0.78** | 0.58 | 5.52 | 0.45 | 0.56 | 0.32 | 5.99 | 0.43 | 0.55 | 0.32 |
| Ours* (Progress Inference) | **3.18** | **0.68** | 0.77 | 0.58 | **5.41** | **0.47** | **0.59** | **0.34** | **5.67** | **0.48** | **0.59** | **0.35** |

## SUPPLEMENTARY MATERIALS

### COMPARISON WITH PRIOR ART WITHOUT BEAM SEARCH

We provide the comparison with state of the arts without using beam search. The results are shown in Table 3. We can see that our proposed method outperformed existing approaches with a large margin on both validation unseen and test sets. Our method with greedy decoding for action selection improved the SR by 9% and 8% on validation unseen and test set. When using progress inference for action selection, the performance on the test set significantly improved by 5% compared to using greedy decoding, yielding 13% improvement over the best existing approach.

### IMPLEMENTATION DETAILS

**Image feature.** Similar to previous work, we use the pre-trained ResNet-152 on ImageNet to extract image features. Each image feature is thus a 2048-d vector. The embedded feature vector for each navigable direction is obtained by concatenating an appearance feature with a 4-d orientation feature $[sin\phi; cos\phi; sin\theta; cos\theta]$, where $\phi$ and $\theta$ are the heading and elevation angles. Following the work in Fried et al. (2018), the 4-dim orientation features are tiled 32 times, resulting a embedding feature vector with 2176 dimension.

**Network architecture.** The embedding dimension for encoding the navigation instruction is 256. We use a dropout layer with ratio 0.5 after the embedding layer. We then encode the instruction using a regular LSTM, and the hidden state is 512 dimensional. The MLP $g$ used for projecting the raw image feature is $BN \rightarrow FC \rightarrow BN \rightarrow Dropout \rightarrow ReLU$. The FC layer projects the 2176-d input vector to a 1024-d vector, and the dropout ratio is set to be 0.5. The hidden state of the LSTM used for carrying the textual and visual information through time in Eq. 1 is 512. We set the maximum length of instruction to be 80, thus the dimension of the attention weights of textual grounding $\alpha_t$ is also 80. The dimension of the learnable matrices from Eq. 2 to 5 are: $W_x \in \mathbb{R}^{512 \times 512}$, $W_v \in \mathbb{R}^{512 \times 1024}$, $W_a \in \mathbb{R}^{1024 \times 1024}$, $W_h \in \mathbb{R}^{1536 \times 512}$, and $W_{pm} \in \mathbb{R}^{592 \times 1}$.

**Training.** We use ADAM as the optimizer. The learning rate is $1e - 4$ with batch size of 64 consistently through out all experiments. When using beam search, we set the beam size to be 15. We perform categorical sampling during training for action selection.

### SUBMISSION TO VISION AND LANGUAGE NAVIGATION CHALLENGE

For evaluating our proposed approach on the unseen test set, we participate in the Vision and Language Navigation challenge and submitted our result with the full proposed approach to the test server. We achieved 61% success rate and ranked #1 on the test server at the time of writing.

We follow the submission guidelines, where picking the highest confidence trajectory from multiple trials for each instruction is not permissible. This means that using the beam search for competing and selecting a final trajectory is not allow directly. Similar to the submission from Speaker-Follower (Fried et al., 2018), we record all the viewpoints traversed during the beam search process. The final agent traverses through all recorded trajectories by first reaching the end of one trajectory and backtracking to the shared viewpoint with the next trajectory. This means that the agent could backtrack to the start point during this process. The trajectories are however logged according to the

closest previous trajectory, so that when a single agent traverses through all recorded trajectories, the overhead for switching from one trajectory to another can be reduced significantly. The final selected trajectory from beam search is then lastly logged to the trajectory. This therefore yields exactly the same success rate and navigation error, as the metrics are computed according to the last viewpoint from a trajectory.

## QUALITATIVE RESULTS

We provide and discuss additional qualitative results on the *self-monitoring* agent navigating on seen and unseen environments. We first discuss four successful examples in Fig. 5 and 6, and followed by two failure examples in Fig. 7.

## SUCCESSFUL EXAMPLES

In Fig. 5 (a), at the beginning, the agent mostly focuses on "*walk up*" for making the first movement. While the agent keeps its attention on "*walk up*" as completed instruction or ongoing action, it shifts the attention on instruction to "*turn right*" as it walks up the stairs. Once it reached the top of the stairs, it decides to turn right according to the grounded instruction. Once turned right, we can again see that the agent pays attention on both the past action "*turn right*" and next action "*walk straight to bedroom*". The agent continues to do so until it decides to stop by grounding on the word "*stop*".

In Fig. 5 (b), the agent starts by focusing on both "*enter bedroom from balcony*" and "*turn left*" to navigate. It correctly shifts the attention on textual grounding on the following instruction. Interestingly, the given instruction "*walk straight across rug to room*" at step 3 is ambiguous since there are two rooms across the rug. Our agent decided to sneak out of the first room on the left and noticed that it does not match with the description from instruction. It then moved to another room across the rug and decided to stop because there is a rug inside the room as described.

In Fig. 6 (a), the given instruction is ambiguous as it only asks the agent to take actions around the stairs. Since there are multiple duplicated actions described in the instruction, e.g. "*walk up*" and "*turn left*", only an agent that is able to precisely follow the instruction step-by-step can successfully complete the task. Otherwise, the agent is likely to stop early before it reaches the goal. The agent also needs to demonstrate its ability to assess the completeness of instruction-following task in order to correctly stop at the right amount of repeated actions as described in the instruction.

In Fig. 6 (b), at the beginning (step 0), the agent only focuses on 'left' for making the first movement (the agent is originally facing the painting). We can see that at each step, the agent correctly focuses on parts of the instruction for making every movements, and it finally believes that the instruction is completed (attention on the last sentence period) and stopped.

## FAILURE EXAMPLES

In Fig. 7 (a) step 1, although the attention on instruction correctly focused on "*take a left*" and "*go down*", the agent failed to follow the instruction and was not able to complete the task. We can however see that the progress monitor correctly reflected that the agent did not follow the given instruction successfully. The agent ended up stopping with progress monitor reporting that only 16% of the instruction was completed.

In Fig. 7 (b) step 2, the attention on instruction only focuses on "*go down*" and thus failed to associate the "*go down steps*" with the stairs previously mentioned in "*turn right to stairs*". The agent was however able to follow the rest of the instruction correctly by turning right and stopping near a mirror. Note that, different from Fig. 7 (a), the final estimated completeness of instruction-following from progress monitor is much higher (16%), which indicates that the agent failed to be aware that it was not correctly following the instruction.

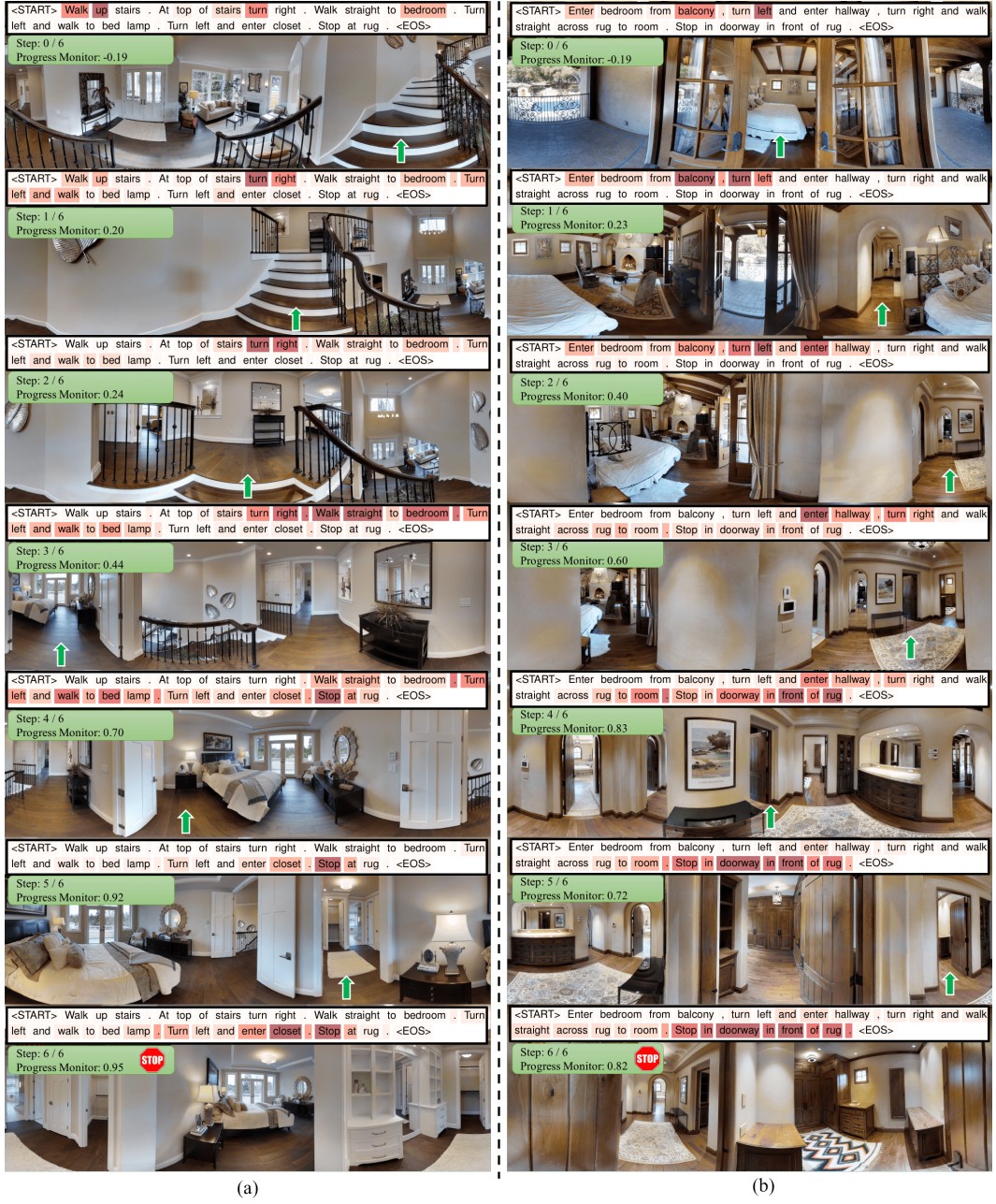

Figure 5: Successful self-monitoring agent navigates in two different unseen environments. Given the navigational instruction located at the top of the figure, the agent starts from starting position and follows the instruction towards the goal. The percentage of instruction completeness estimated by the proposed progress monitor gradually increases as the agent navigates and approaches the goal.

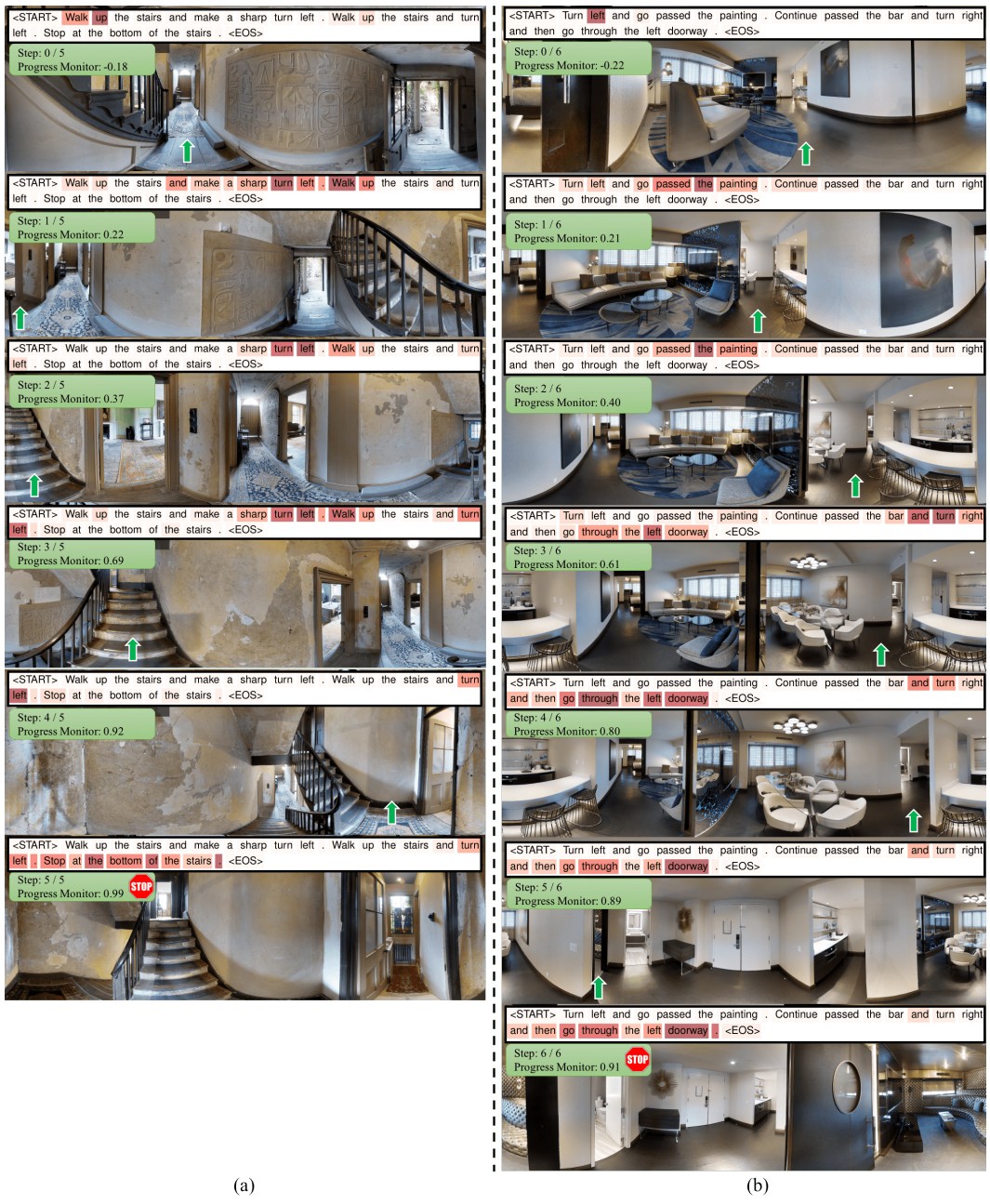

(a)                                                                (b)

Figure 6: Successful self-monitoring agent navigates in (a) unseen and (b) seen environments. (a) The given instruction is ambiguous as it only asks the agent to take actions around the stairs. Since there are multiple duplicated actions described in the instruction, e.g. "*walk up*" and "*turn left*", only an agent that is able to precisely follow the instruction step-by-step can successfully complete the task. Otherwise, the agent is likely to stop early before it reaches the goal. (b) The agent correctly pays attention to parts of the instruction for making decisions on selecting navigable directions. Both the agents decide to stop when shifting the textual grounding on the last sentence period.

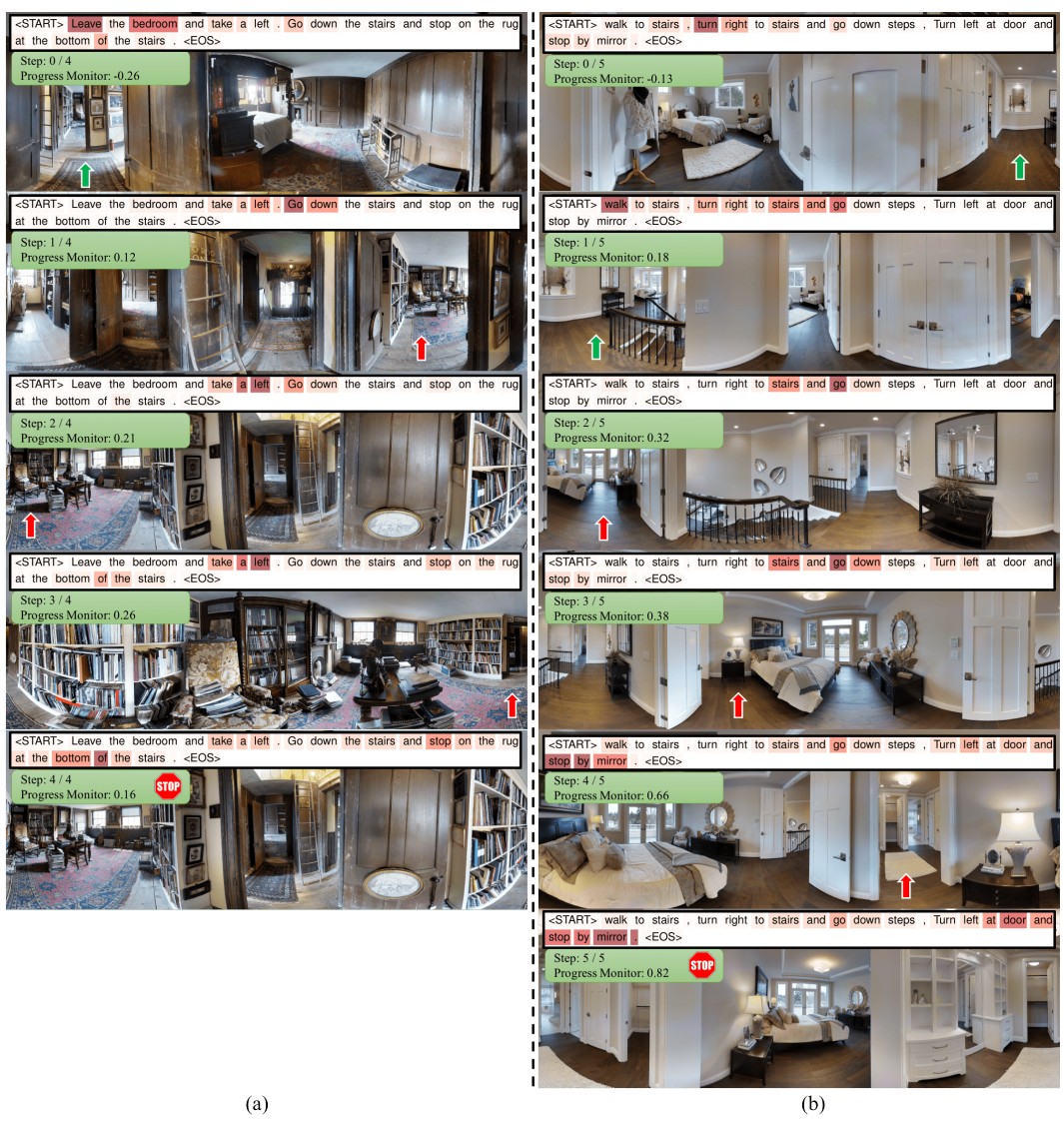

(a)                                    (b)

Figure 7: Failed self-monitoring agent navigates in unseen environments. (a) The agent missed the "*take a left*" at step 1, and consequently unable to follow the following instruction correctly. However, note that the progress monitor correctly reflected that the instruction was not completed. When the agent decides to end the navigation, it reports that only 16% of the instruction was completed. (b) At step 2, the attention on instruction only focuses on "*go down*" and thus failed to associate the "*go down steps*" with the stairs previously mentioned in "*turn right to stairs*". The agent was however able to follow the rest of the instruction correctly by turning right and stopping near a mirror. Note that, different from (a), the final estimated completeness of instruction-following is much higher, which suggests that the agent failed to correctly be aware of its progress towards the goal.

