# OpenReview forum: "Self-Monitoring Navigation Agent via Auxiliary Progress Estimation"
_ICLR.cc/2019/Conference_

### Official Review · AnonReviewer2 · 2018-10-29

**Rating:** 7
**Confidence:** 4

**Review:**

This paper describes a model for vision-and-language navigation. The proposed
model adds two components to the baseline model proposed by Fried et al. (2018):

- a panoramic visual attention (referred to in this paper as "visual--textual
  co-grounding"), in which the full scene around the agent's current position is
  attended to prior to selecting a direction to follow

- an auxiliary "progress monitoring" loss which encourages the agent to to
  produce textual attentions from which the distance to the goal can be directly
  inferred

The two components combine to give state-of-the-art results on the Room2Room
dataset: small improvements over existing approaches on the "-seen" evaluation
set and larger improvements on the "-unseen" evaluation sets. These improvements
also stack with the data-augmentation approach of Fried et al.

I think this is a reasonable submission and should probably be accepted. However, I
have some concerns about presentation and a number of specific questions about
model implementation and evaluation.

PRESENTATION AND NAMING

First off: I implore the authors to find some descriptor other than "self-aware"
for the proposed model. "Self-aware" is an imprecise description of the agent in
this paper---the agent is specifically "aware" of its visual surroundings and
its distance from the goal, neither of which is meaningfully an aspect of
"self". Moreover, self-awareness means something quite different in adjacent
areas of cognitive science and philosophy; overloading the term in the specific
(and comparatively mundane) way used here creates confusion. See section 3.4 of
https://arxiv.org/abs/1807.03341 for broader discussion. Perhaps something
like "visual / temporal context-sensitivity" to describe what's new here? A bit
clunky, but I think it makes the contributions of this work much clearer.

As suggested in the summary above, I also think "visual--textual co-attention"
is also an unhelpfully vague description of this aspect of the contribution. The
textual attention mechanism used in this paper is the same as in all previous
work on the task. Representations of language don't even interact with the
visual attention mechanism except by way of the hidden state, and the salient
new feature of the visual attention is the fact that it considers the full
panoramic context before choosing a direction.

MODELING QUESTIONS

- p4: $y_t^{pm}$ is defined as the "normalized distance from the current
  viewpoint to the goal". Is this distance in units of length (as defined by the
  simulator) or units of time (i.e. the number of discrete "steps" needed to
  reach the goal)?

  The authors have already clarified on OpenReview that the progress monitor
  objective uses an MSE loss rather than a likelihood loss. Do I understand
  correctly that ground-truth distances are in [0, 1] but model predictions are
  in [-1, 1]? Why not use a sigmoid? Also, how does scoring beam-search
  candidates as $p_t^{pm} \times p_{k,t}$ work if $p_t^{pm}$ can flip the sign?

- The input to the progress monitor is formed by concatenating the attention
  vector $\alpha_t$ to a vector of state features, and then multiplying by a
  fixed weight matrix. How is this possible? The size of $\alpha_t$ varies
  depending on the length of the instruction sequence. Are attentions padded out
  to the length of the longest instruction in the training set? If so, how can
  the model learn when it's reached the end of a short instruction sequence?
  What would happen if the agent encountered a sequence that was too long?

EVALUATION QUESTIONS

- The progress monitor is used both as an auxiliary training objective and as a
  beam search heuristic. Is it possible to disentangle these two contributions?
  (E.g. by ignoring the scores during beam search, or by doing augmented beam
  search in a model that was trained without the auxiliary objective.)

- Not critical, but it would be nice to know if the contributions here stack
  with the pragmatic inference procedure in Fried et al.

- While, as pointed out on OpenReview, it is not required to include SPL
  evaluations, I think it would be informative to do so---the preliminary
  results with no beam search look good!

MISCELLANEOUS

p1: "without a map" If you can do beam search, you effectively have a map.

p1: "...smoothly" What does "smoothly" mean in this context?

p2: "the position of grounded instruction can follow past and future
    instructions". Is the claim here that if instructions are of the form "ACB"
    and the agent is supposed to do "ABC", that the proposed model will execute
    these instructions successfully and the baseline will not? This claim does
    not appear to be evaluated anywhere in the body of the paper.

p4: "intelligently prunes" "Intelligently" is unnecessary.

p4: "for empirical reasons" What does this mean?

p5: "Intuitively, an instruction-following agent is required..." The existence
    of non-attentive models that do reasonably well at these
    instruction-following tasks suggest that this is not actually a requirement.

---

> ### Author Response · Authors · 2018-11-21
> **Results for sigmoid, ablation study table updated, SPL metric included.**
>
> Hi,
>
> We would like to thank the reviewer for their thoughtful and constructive feedback.
>
> 1. Regarding “presentation and naming”:
>
> We thank the reviewer for pointing out the potential issue regarding the naming of the paper. We agree with the reviewer and we have been trying our best to find a better name. For now, one of the options that we come up with is “Self-Monitoring Navigation Agent via Auxiliary Progress Estimation”. We hope this new title is more clear and suitable for the work. If the reviewers have other suggestions, please kindly let us know.
>
> 2. Regarding “the definition of distance”:
>
> The distance is defined in units of length the same as the simulator. We also agree that using the number of steps is also a reasonable approach, and it would be interesting to explore in the future. We have clarified the definition in the revision.
>
> 3. Regarding ”using sigmoid as opposed to tanh”:
>
> We have found that using sigmoid performs similarly to using the tanh function. The results on different metrics are shown below:
> -----------------------------------------------------------------------------------------------------------------------------------
>                          	   Val-seen	                    	   Val-unseen
>                                    NE↓    SR↑    OSR↑    SPL↑     NE↓    SR↑    OSR↑    SPL↑
> -----------------------------------------------------------------------------------------------------------------------------------
> Ours (Tanh)		   3.72    0.63    0.75     0.56     5.98    0.44    0.58      0.30
> Ours (Sigmoid)        3.72    0.64    0.72     0.59	   5.92    0.44    0.56      0.33
> -----------------------------------------------------------------------------------------------------------------------------------
>
> The main incentive for using tanh is to allow the agent to wander around when it’s distance to the goal is larger than the original starting distance. As a result, this allows the progress monitor to output values lower than 0 indicating that the agent has lost the track of textual grounding.
>
> We simply normalize the output of progress monitor between 0 to 1 before combining the score for beam search. We have revised the paper to clarify this accordingly.
>
> 4. Regarding “instructions with various lengths”:
>
> We use zero-padding to handle instructions with various lengths. We have also explored ideas like using interpolation to upsample the attention weights of short instructions to a fixed length of 80, but it produces a similar performance as without interpolation.
> Generally, we have observed that the validation samples with longer instructions are more likely to fail, but this may due to the fact that the required total number of steps of these samples is larger. Hence, the agent is prone to errors since it needs to predict actions correctly for more steps.

---

> > ### Author Response · Authors · 2018-11-21
> > **(Cont’d)**
> >
> >
> > 5. Regarding “using beam search without auxiliary objective”:
> >
> > Yes, and we have provided the result of using beam search with the co-grounding model (without progress monitor) in the updated ablation study table as shown below. We have also included this updated ablation study table in the revision.
> >
> > ------------------------------------------------------------------------------------------------------------------------------------------------------------------------
> > 				                     Inference Mode			                                Validation-Seen           Validation-Unseen
> > 	       Co-	              Progress      Greedy      Progress     Beam	        Data
> > 	#    Grounding    Monitor      Decoding   Inference    Search	Aug.	NE↓  SR↑  OSR↑  SPL↑     NE↓       SR↑  OSR↑  SPL↑
> > ------------------------------------------------------------------------------------------------------------------------------------------------------------------------
> > Baseline 								                                                        4.36  0.54  0.68       -       7.22        0.27  0.39     -
> > ------------------------------------------------------------------------------------------------------------------------------------------------------------------------
> > 	        1      ✔ 			                  ✔				                                        3.65  0.65  0.75    0.56    6.07 	0.42   0.57    0.28
> > Ours	2      ✔ 	            ✔ 		  ✔				                                        3.72  0.63  0.75    0.56    5.98 	0.44   0.58    0.30
> > 	        3      ✔	            ✔		          ✔			                                   ✔ 	        3.22  0.67  0.78    0.58    5.52       0.45   0.56    0.32
> > ------------------------------------------------------------------------------------------------------------------------------------------------------------------------
> > Ours	4      ✔ 	            ✔ 			              ✔			                        3.56  0.65  0.75    0.58    5.89 	0.46   0.60    0.32
> > 	        5      ✔ 	            ✔ 			              ✔		                   ✔	        3.18  0.68  0.77    0.58    5.41 	0.47   0.59    0.34
> > ------------------------------------------------------------------------------------------------------------------------------------------------------------------------
> > 	        6      ✔ 	             				                             ✔		                3.66  0.66  0.76    0.62    5.70 	0.49   0.68    0.42
> > Ours	7      ✔ 	            ✔ 				                     ✔		                3.23  0.70  0.78    0.66    5.04 	0.57   0.70    0.51
> > 	        8      ✔ 	            ✔ 				                     ✔	           ✔	        3.04  0.71  0.78    0.67    4.62 	0.58   0.68    0.52
> > ------------------------------------------------------------------------------------------------------------------------------------------------------------------------
> >
> > 6. Regarding “if contribution stacks with the pragmatic inference”:
> >
> > We believe that the proposed method would stack with the Pragmatic Inference in Fried et al with some performance improvement since the underlying methods for “ranking” the candidate routes are different. Pragmatic Inference ranks the routes by learning a mapping from a complete sequence of visual inputs to textual output, whereas the Progress Monitor learns a mapping from incomplete visual-textual grounding output and actions to distance. We expect there will be performance improvement by further using pragmatic inference since these two methods are orthogonal.
> >
> > 7. Regarding “adding SPL metric”:
> >
> > We completely agree with the reviewers that reporting SPL will be beneficial for future research work along this direction. As suggested, the new SPL metric is added along with ALL settings in the updated ablation study table and the original table 1 for comparing with state of the arts as well.
> >
> > 8. Regarding the comments in MISCELLANEOUS:
> >
> > We thank the reviewer for paying extra attention to the details of the paper. We have revised the paper according to the suggestions in the MISCELLANEOUS section and clarified some confusion by rephrasing some sentences. We now provide answers to some of the questions asked.
> >
> > 9. Regarding “the position of grounded instruction can follow past and future instructions”:
> >
> > We meant to indicate that (also as we demonstrated from qualitative results), the positions of grounded instructions at each step reflects the action that is required at this step and the action that was completed from last or previous steps (see Figure 5 and 6 for examples).
> > Also, to the best of our knowledge, the order of the instruction will be the same as the order of actions required from the agent because ground-truth actions are computed from the shortest path on the navigation graph.
> >
> > 10. Regarding “for empirical reasons”:
> >
> > We empirically found that using concatenation leads to slightly higher performance and stable training over using element-wise addition. We have made the change in the revision to make this clear.

---

> > > ### Author Response · Authors · 2018-11-21
> > > **(Cont’d)**
> > >
> > >
> > > 11. Regarding “the existence of non-attentive models”:
> > >
> > > To the best of our knowledge, all existing methods use attention models on the VLN task, but we have shown in Figure 3 that the baseline method (using attention on textual grounding) was not able to successfully track the instruction. Our proposed progress monitor made this possible and demonstrated superior performance across difference metrics.

---

> > > > ### Comment · AnonReviewer2 · 2018-11-29
> > > > **update**
> > > >
> > > > Dear authors,
> > > >
> > > > Thanks for the detailed response! The new revision addresses my presentation concerns and answers my questions, so I've increased my score to a 7. I still think it would be useful to present a slightly more targeted set of ablations in the main paper rather than the kitchen sink in table 2: e.g. just
> > > > - "co-grounding" / nothing
> > > > - progress monitor / progress inference / both / neither
> > > > - best model + data augmentation / best model (no data aug)
> > > >
> > > > Haven't proofread the new draft carefully but "state of the arts" in Table 1 is wrong, so you should do another copyediting pass before the final version if the paper is accepted.

---

### Official Review · AnonReviewer1 · 2018-11-02
**Good idea, unclear results**

**Rating:** 6
**Confidence:** 4

**Review:**

This submission introduces a new method for vision+language navigation which tracks progress on the instruction using a progress monitor and a visual-textual co-grounding module. The method is shown to perform well on a standard benchmark. Ablation tests indicate the importance of each component of the model. Qualitative examples show that the proposed method attends to different parts of the instruction as the agent moves.

Here are some comments/questions:
- I like the underlying idea behind the method. The manuscript is written well for most parts.
- The qualitative examples and Figure 2 are really helpful in understanding the reasons behind the improved performance.
- There is a lot of confusion regarding the use of beam search. It's unclear from the current manuscript which results are with and without beam search. It seems like beam search was added from Ours 1 to Ours 2 in Table 2. It's not clear which rows involve beam search in Table 1. Some concerns about beam width were raised in the comments which I agree with. Please modify the submission to clearly indicate the use of beam search for each result and specify the beam width.
- The use of beam search seems unrealistic to me as I can not think of any way a navigational model using beam search can be transferred or applied to real-world. I understand that one of the baselines uses beam search, so it's fair for performance comparison purposes, but could you provide any justification of how it might be useful in real-world? If there's no reasonable justification, could you also provide all the results (along with SPL metric) without beam search, including ablation, comparing with only methods without beam search?
- I do not understand why the OSR in the submission is 0.64 and 0.70 for Speaker-Follower and proposed method and 0.96 and 0.97 in the comments.
- It seems like the proposed method is tailored for the VLN task. In many real-world scenarios, an agent might be given an instruction which only describes the goal (such as in Chaplot et al. 2017 and Hermann et al. 2017) and not the path to the goal, could the authors provide their thoughts on whether the proposed would work well for such instructions? What would the progress monitor and textual attention distribution learn in such a scenario?

Due to confusion about results and concerns about beam search, I give a rating of 5. I am willing to increase the rating if the authors address the above concerns.

---

> ### Author Response · Authors · 2018-11-21
> **SOTA and ablation study tables updated, SPL metric included, comparison without beam search added**
>
> Hi,
>
> We would like to thank the reviewer for the thoughtful and constructive feedback.
>
> 1. Regarding “confusion regarding the result of beam search”:
>
> The updated ablation study table is shown below, and the same table has been added to the revised paper. In this table, we show the performance improvement of each component with different inference modes (which are described in the revised paper). We have also added the recently introduced SPL metric under all settings to table. From the results, we can again see that the proposed components improved the performance across different evaluation metrics.
>
> ------------------------------------------------------------------------------------------------------------------------------------------------------------------------
> 				                     Inference Mode			                                Validation-Seen           Validation-Unseen
> 	       Co-	              Progress      Greedy      Progress     Beam	        Data
> 	#    Grounding    Monitor      Decoding   Inference    Search	Aug.	NE↓  SR↑  OSR↑  SPL↑     NE↓       SR↑  OSR↑  SPL↑
> ------------------------------------------------------------------------------------------------------------------------------------------------------------------------
> Baseline 								                                                        4.36  0.54  0.68       -       7.22        0.27  0.39       -
> ------------------------------------------------------------------------------------------------------------------------------------------------------------------------
> 	        1      ✔ 			                  ✔				                                        3.65  0.65  0.75    0.56    6.07 	0.42   0.57    0.28
> Ours	2      ✔ 	            ✔ 		  ✔				                                        3.72  0.63  0.75    0.56    5.98 	0.44   0.58    0.30
> 	        3      ✔	            ✔		          ✔			                                   ✔ 	        3.22  0.67  0.78    0.58    5.52       0.45   0.56    0.32
> ------------------------------------------------------------------------------------------------------------------------------------------------------------------------
> Ours	4      ✔ 	            ✔ 			              ✔			                        3.56  0.65  0.75    0.58    5.89 	0.46   0.60    0.32
> 	        5      ✔ 	            ✔ 			              ✔		                   ✔	        3.18  0.68  0.77    0.58    5.41 	0.47   0.59    0.34
> ------------------------------------------------------------------------------------------------------------------------------------------------------------------------
> 	        6      ✔ 	             				                             ✔		                3.66  0.66  0.76    0.62    5.70 	0.49   0.68    0.42
> Ours	7      ✔ 	            ✔ 				                     ✔		                3.23  0.70  0.78    0.66    5.04 	0.57   0.70    0.51
> 	        8      ✔ 	            ✔ 				                     ✔	           ✔	        3.04  0.71  0.78    0.67    4.62 	0.58   0.68    0.52
> ------------------------------------------------------------------------------------------------------------------------------------------------------------------------
>
> 2. Regarding “The use of beam search and SPL metric”:
>
> We have provided all the results without beam search in the updated ablation study table (see table above or from the revised paper), and the comparison with methods without beam search is shown in the table below (for Speaker-Follower, the Pragmatic Inference which relies on beam search is removed for comparison purpose). This table is also included in the revised paper.

---

> > ### Author Response · Authors · 2018-11-21
> > **(Cont’d)**
> >
> >
> > ------------------------------------------------------------------------------------------------------------------------------------------------------------
> > 	                                            Val-seen	   	                    Val-unseen		                  Test
> >                                                     NE↓    SR↑    OSR↑    SPL↑	    NE↓    SR↑      OSR↑    SPL↑         NE↓    SR↑    OSR↑    SPL↑
> > ------------------------------------------------------------------------------------------------------------------------------------------------------------
> > Student-forcing		           6.01    0.39    0.53      -	    7.81    0.22    0.28       -	           7.85    0.20    0.27       0.18
> > RPA			                           5.56    0.43    0.53      -	    7.65    0.25    0.32       -	           7.53    0.25    0.33       0.23
> > Speaker-Follower		           3.36    0.66    0.74      -	    6.62    0.36    0.45       -	           6.62    0.35    0.44       0.28
> > ------------------------------------------------------------------------------------------------------------------------------------------------------------
> > Ours (Greedy Decoding)	   3.22    0.67    0.78      0.58	    5.52    0.45    0.56       0.32	   5.99    0.43    0.55       0.32
> > Ours (Progress Inference)	   3.18    0.68    0.77      0.58	    5.41    0.47    0.59       0.34	   5.67    0.48    0.59       0.35
> > ------------------------------------------------------------------------------------------------------------------------------------------------------------
> >
> > In addition to the comparison without beam search provided above, we would like to elaborate on the reason beam search is important given the research progress in this direction.
> >
> > Admittedly, beam search exhaustively searches a much larger space so that the agent can better decide which direction to go and when to stop. The VLN task along with the R2R dataset was recently introduced less than 1 year ago with a more than 60% SR gap between the best-known model and human performance. Each work including ours has step by step brought this gap down to 25%. We agree that in the long-term, ideally, the common goal of the research community is to develop an agent that achieves a high success rate while maintaining a low trajectory length. We argue that given the complexity of the VLN task which requires the agent to simultaneously achieve visual grounding, textual reasoning, temporal memorization/reasoning, and intelligent action selection to navigate, the need to relax the task along multiple directions in order to make progress is important and essential. There is no current best model for both metrics (SR and SPL), and beam search typically differentiates the two regimes. Also, whether the beam search is realistic or not depends on the application. For example, in robotics, it is not atypical to have some exploration and/or mapping of the environment as well, after which beam search can be utilized. In fairness and future comparison, we have provided SR and SPL metrics for our proposed method and all settings in the ablation study table.
> >
> > 3. Regarding “OSR differences in the submission and OpenReview comment”:
> >
> > The OSR were 0.96 and 0.97 respectively due to the fact that, when submitting the results to the test server, it is strictly required that all submissions need to include all locations that the agent traversed. Since beam search explores the environment, the trajectory length is usually significantly higher. The chance of passing/reaching the goal is also higher, hence the OSR is close to 1. In order to be consistent with how the existing work reports their performance (see the recently updated Speaker-Follower paper for example), we follow the same convention: when using beam search, only the performance on the test set includes all viewpoints traversed. The performance reported on the validation set use only the highest ranked trajectory after beam search.
> >
> > 4. Regarding “scenarios of instructions describe the goal, not the path”:
> >
> > Our proposed agent learns to infer and leverage the progress monitor to constrain and regularize the textual grounding module. We believe that the high-level concept of the progress monitor will work as long as inferring progress made towards the goal can be done for the given task, i.e. there is some information in the grounding or visual information to accurately estimate it. Using textual attention distribution is just one instantiation that we explore on leveraging the progress monitor for the VLN task.

---

> > > ### Author Response · Authors · 2018-11-21
> > > **(Cont’d)**
> > >
> > >
> > > If the goal is given then appearance features can inform progress, both immediately near the goal but also contextually (e.g. rooms that tend to be near the goal or tend to contain the object in the goal). In a scenario where the instruction only describes the goal rather than the path to the goal, the agent will require a map or positions to estimate the progress using the proposed progress monitor. It would be interesting to see when the progress monitor combines with the semantic maps proposed recently in [1], [2], or [3], where the positions of the semantic map are also associated with the progress prediction condition on the given instruction. The agent can be constrained to select directions that have the closest image representation with the expected image feature representation extracted from semantic map representation and use the associated progress estimate as an additional indicator for action selection.
> > >
> > > [1] Gordon, Daniel, et al. "IQA: Visual question answering in interactive environments." CVPR. 2018.
> > > [2] Savinov, Nikolay, Alexey Dosovitskiy, and Vladlen Koltun. "Semi-parametric topological memory for navigation." ICLR (2018).
> > > [3] Walter, Matthew R., et al. "A framework for learning semantic maps from grounded natural language descriptions." The International Journal of Robotics Research 33.9 (2014): 1167-1190.

---

> > > > ### Comment · AnonReviewer1 · 2018-11-30
> > > > **Updating score**
> > > >
> > > > Most of my questions were answered by the authors. I raise my rating to 6 based on the response and revised paper. However, I am not convinced by the argument for using beam search. I do not believe something unrealistic should be used to get better numbers just because the task is challenging. The authors mention that beam search can be used in robotics but do not provide any reference. I highly doubt the effectiveness of beam-search in an imperfect mapping of an environment, with realistic fine-grained motion as compared to discretized motion and perfect forward model in the simulated environment used. Nevertheless, the proposed model provides better performance even without beam search.

---

### Official Review · AnonReviewer3 · 2018-11-05
**Interesting Approach to Route Instruction Following with Thorough Evaluation**

**Rating:** 8
**Confidence:** 5

**Review:**

The paper considers the problem of following natural language route instructions in an unknown environment given only images. Integral to the proposed ("self-aware") approach is its ability to reason over which aspects of the instruction have been completed, which are to be followed next, which direction to go in next, as well as the agents current progress. This involves two primary components of the architecture. The first is a visual-textual module that grounds to the completed instruction, the next instruction, and the next direction based upon the visual input. The second is a "progress monitor" that takes the grounded instruction as input and captures the agent's progress towards completing the instruction.


STRENGTHS

+ The paper describes an interesting approach to reasoning over which aspects of a given instruction have been correctly followed and which aspect to act on next. This takes the form of a visual-textual co-grounding model that identifies the instruction previously completed, the instruction corresponding to the next action, and the subsequent direction in which to move. The inclusion of a "progress monitor" allows the method to reason over whether the navigational progress matches the instruction.

+ The paper provides a thorough evaluation on a challenging benchmark language understanding dataset. This evaluation includes detailed comparisons to state-of-the-art baselines together with ablation studies to understand the contribution of the different components of the architecture.

+ The paper is well written and provides a thorough description of the framework with sufficient details to support replication of the results.


WEAKNESSES

- The paper would benefit from a more compelling argument for the importance of reasoning over which aspects of the instruction have been completed vs. which to act on next.

- The paper emphasizes the use of images, the visual grounding reasons over visual features.

- The paper incorrectly states that existing methods for language understanding require an explicit representation of the target. Several existing methods do not have this requirement. For example, Matuszek et al., 2012 parse free-form language into a formal logic representation for a downstream controller that interprets these instructions in unknown environments. Meanwhile, Duvallet et al., 2014 and Hemachandra et al., 2015 exploit language (together with vision and LIDAR) to learn a distribution over the unknown environment that guides grounding. Meanwhile, Mei et al., 2016 reason only over natural language text and parsed images, without knowledge of the environment or an explicit representation of the goal.

C. Matuszek, E. Herbst, L. Zettlemoyer, and D. Fox, “Learning to parse natural language commands to a robot control system,” in Proceedings of the International Symposium on Experimental Robotics (ISER), 2012.

S. Hemachandra, F. Duvallet, T. M. Howard, N. Roy, A. Stentz, and M. R. Walter, “Learning models for following natural language directions in unknown environments,” in Proc. IEEE Int’l Conf. on Robotics and Automation (ICRA), 2015

F. Duvallet, M. R. Walter, T. Howard, S. Hemachandra, J. Oh, S. Teller, N. Roy, and A. Stentz, “Inferring maps and behaviors
from natural language instructions,” in Proceedings of the International Symposium on Experimental Robotics (ISER), 2014.

- While it's not a neural approach, the work of Arkin et al., 2017 which reasons over the entire instruction history when deciding on actions (through a statistical symbol grounding formulation)⁠

J. Arkin, M. Walter, A. Boteanu, M. Napoli, H. Biggie, H. Kress-Gazit, and T. Howard. "Contextual Awareness: Understanding Monologic Natural Language Instructions for Autonomous Robots," In Proceedings of the IEEE International Symposium on Robot and Human Interactive Communication (RO-MAN), 2017

- The paper misses the large body of literature on grounded language acquisition for robotics.

QUESTIONS

* What is the effect of using positional encoding for textual grounding as opposed to standard alignment methods such as those used by Mei et al., 2016?

* Perhaps I missed it, but what happens if instructions are specified in such a way that their ordering is not consistent with the correct action ordering (e.g., with corrections interjected)?

---

> ### Author Response · Authors · 2018-11-22
> **Additional results without positional encoding and related work added.**
>
>
> Hi,
>
> We would like to thank the reviewer for the thoughtful and constructive feedback.
>
> We thank the reviewer for bringing the additional literature from related fields to our attention. We have included and discussed them in the revised paper (please see both introduction and related work sections for the changes we made).
>
> 1. Regarding “importance of reasoning over completed or next instruction”:
>
> We rewrote the sentences in the second paragraph in the introduction and try to make the reason why reasoning over both past and present instructions is important and essential. Please see the revised paper for the changes we made. We emphasize that the transition between past and next part of the instructions is a soft boundary, in order to determine when to transit and to follow the instruction correctly the agent is required to keep track of both grounded instructions.
>
> 2. Regarding “visual grounding over visual features”:
>
> In order to provide a fair comparison with prior arts, we chose to use the image feature vector provided directly with this task. Our current visual grounding module performs attention over different parts of the panoramic image and grounds the located instruction to a part of the panoramic image. To further provide fine-grained visual grounding, we agree that it would be interesting to use panoramic images directly as input and perform visual grounding on feature maps or object-level bounding boxes.
>
> 3. Regarding “effect of positional encoding”:
>
> In our early experiments, we found that, although removing positional encoding can achieve better results on val-seen, the agent overfits quickly on val-unseen. We believe that the agent’s ability to generalize to unseen environments is more important than achieving good results on val-seen. Thus, we use positional encoding for ablation study and produce the final result.
> -----------------------------------------------------------------------------------------------------------------------------------
> 	                            Val-seen	   	                   Val-unseen
>                                     NE↓    SR↑    OSR↑    SPL↑    NE↓     SR↑     OSR↑    SPL↑
> -----------------------------------------------------------------------------------------------------------------------------------
> Ours (No PE)  	    3.37    0.69    0.78     0.61	    6.04    0.42    0.55      0.30
> Ours			    3.72    0.63    0.75     0.56     5.98    0.44    0.58      0.30
> -----------------------------------------------------------------------------------------------------------------------------------
>
> 4. Regarding “what if the orderings of the instruction and actions are inconsistent”:
>
> The Room-to-Room dataset comes with the ground-truth starting location, goal location, and the instruction associated with it. The ground-truth trajectory is computed as the shortest path from the navigational graph from starting location to the goal, and the quality of given instructions is verified by humans which achieved 86% success rate on the test set. From our own observation, the ordering of actions is consistent with the ordering of instructions. If this was not the case, our current language grounding mechanism may still applicable since it can represent an arbitrary weighting over the sentence. Similarly, since progress monitoring is a learned function over this it could still learn to estimate progress. However, note that our assessment depends on how different the orders of instruction and actions are. If the difference is small, our agent is very likely to be able to recover from incorrect instruction. We believe this can be an interesting direction for future work.

---

### Public Comment · (anonymous) · 2018-10-20
**Concerns on the evaluation metric and the so-called SOTA performance**

This paper only reports the absolute Success Rate as the evaluation metric and hides the trajectory lengths. It is well known that the Success Rate can be generally improved by exhaustedly exploring the environment before committing to a decision. However, beam search is not appropriate for robotics, because longer trajectories have more costs (battery, wear, delays for the user, etc).

Therefore, Success rate weighted by normalized inverse Path Length (SPL) trades-off Success Rate against Trajectory Length. SPL is defined in the paper On Evaluation of Embodied Navigation Agents (https://arxiv.org/abs/1807.06757) and introduced as one of the evaluation metrics for the VLN task.

I am not sure why the authors didn't include the trajectory lengths in the paper. But from the VLN challenge leaderboard, the SPL score of the authors' submission is only 0.02 (out of 1.00), which is severely worse than the Seq2Seq baseline (0.18). The trajectory length is 373.09 meters. It seems like the authors are gaming the Success Rate with exhaustive search. Hence, I do not think it is proper to claim that the method has achieved new SOTA performance.

---

> ### Public Comment · ~Peter_Anderson1 · 2018-10-22
> **A note from the challenge organizers on the SPL metric**
>
> ICLR reviewer guidelines state that "no paper will be considered prior work if it appeared on arxiv, or another online venue, less than 30 days prior to the ICLR deadline." The paper defining the SPL metric appeared on arXiv on 18 July. However, as an organizer of the VLN challenge and a co-author of the arXiv paper mentioned above, I would like to state for the benefit of reviewers that the SPL metric was not added to the public VLN leaderboard until September 8th (19 days before the ICLR deadline). In fairness to authors with work in progress, reviewers may wish to exclude this metric from the definition of prior work for ICLR 2019 since it was not implemented on the leaderboard 30 days prior to the deadline. Existing work on the dataset has been primarily evaluated in terms of 'Success Rate', as reported in this submission.

---

> ### Author Response · Authors · 2018-10-23
> **Results without beam search**
>
> Hi,
>
> Thank you for raising an important discussion about metrics.
>
> The VLN task was recently introduced less than 1 year ago with a more than 60% success rate gap between the best-known model and human performance. Each existing work has step by step helped us to advance and reduce this gap to 33%.
>
> Our proposed work, when combined with beam search, is able to further close the gap to 25% measured with success rate. We focused on this metric since that was the metric used by recent state of art. Yet, there is obviously still room for improvement. Ideally, the common goal from the research community is to develop an agent that achieves a high success rate with low trajectory length. We argue that given the complexity of the VLN task which requires the agent to simultaneously achieve visual grounding, textual reasoning, temporal memorization/reasoning, and intelligently select actions to navigate, the need to relax the task along multiple directions in order to make progress is important and essential. There is no current best model for both metrics (SR and SPL), and beam search typically differentiates the two regimes.
>
> Even from the robotics perspective, these are two important objectives that one might want to trade off (length/time and success rate) given that there is no one solution that is pareto-optimal, and beam search can be seen as an exploration mechanism which is not uncommon in robotics. Beam search, which thoroughly explores the environment, eases the burden for the agent in intelligently selecting actions given the progress made towards the goal so that the agent can solely focus on identifying the implicit target represented by navigational instruction. However, the best performing model with beam search is still more than 20% lower than the human performance. Further, note that our state of art success rate has only about 29% of the trajectory length of the Speaker-Follower model which also uses beam search.
>
> Nonetheless, we agree that the newly introduced SPL metric is also important, though it emphasizes a different aspect of the navigation task. For future comparison, we thus submitted our proposed model without beam search to the test server. Our result achieves state of art SPL results compared to existing approaches (note that we exclude submissions after the ICLR deadline) and is shown in the table below (the leaderboard only allows one result to be shown from a team). For each metric, with or without beam search, our proposed method outperforms existing approaches by a large margin.
>
> --------------------------------------------------------------------------------------------------------------------------------------
> 	  	      						               Test-Unseen
>    							length↓    NE↓     SR↑     OSR↑  	SPL↑
> --------------------------------------------------------------------------------------------------------------------------------------
> 									without beam search
> --------------------------------------------------------------------------------------------------------------------------------------
> Seq2Seq Baseline		   	8.13           7.85     0.20    0.27       0.18
> Look Before You Leap	   	        9.15	          7.53     0.25    0.32       0.23
> Ours		  	 			18.04         5.67     0.48    0.59       0.35
> --------------------------------------------------------------------------------------------------------------------------------------
> 									with beam search
> --------------------------------------------------------------------------------------------------------------------------------------
> Speaker-Follower		   	         1257.38    4.87     0.53    0.96   	0.01
> Ours			   			 373.09      4.48     0.61    0.97   	0.02
> --------------------------------------------------------------------------------------------------------------------------------------

---

> > ### Public Comment · (anonymous) · 2018-10-25
> > **Consider adding ablations to paper / supplementary?**
> >
> > I think the ablation with or without beam search is very valuable, please add it to paper or at least supplementary?
> > Plus, ablation with (only visual grounding) (only textual grounding) (only cogrounding without progress monitor) would be very illuminating as well.

---

> > > ### Author Response · Authors · 2018-10-28
> > > **Additional results for ablation study**
> > >
> > > Hi,
> > >
> > > Thank you for the suggestions on ablation study. Below are the results as requested. We replaced the soft attention on visual or textual inputs with a simple mean pooling, e.g., only visual grounding means we simply use mean-pooling on textual input, and vice versa.
> > >
> > >
> > > ------------------------------------------------------------------------------------------------------------------------------------------------
> > > 			   Co-Grounding	  Progress    Beam 	  Validataion-Seen 	        Validation-Unseen
> > > 		    #     Visual	Textual     Monitor     Search 	  NE↓        SR↑        OSR↑ 	NE↓	         SR↑        OSR↑
> > > ------------------------------------------------------------------------------------------------------------------------------------------------
> > > Baseline 										  4.36       0.54 	0.68	        7.22         0.27        0.39
> > > ------------------------------------------------------------------------------------------------------------------------------------------------
> > > 		    1         ✔ 								  3.94 	0.62 	0.73 	6.34 	0.40 	0.53
> > > 		    2    	             ✔						  3.60 	0.65 	0.75 	6.27 	0.43 	0.54
> > > Ours	    3         ✔ 	     ✔						  3.65 	0.65 	0.75 	6.07 	0.42 	0.57
> > > 		    4         ✔ 	     ✔		✔ 				  3.56 	0.65 	0.75	        5.89 	0.46 	0.60
> > > 		    5         ✔  	     ✔		✔ 		  ✔ 		  3.23 	0.70 	0.78 	5.04 	0.57 	0.70
> > > ------------------------------------------------------------------------------------------------------------------------------------------------

---

> > > > ### Public Comment · (anonymous) · 2018-10-28
> > > > **Beam size**
> > > >
> > > > thanks for the ablation study.
> > > >
> > > > I notice that in your paper the beam size is 15, while the beam size used in the speaker-follower model is 40. Playing with the beam size can influence the performance and the trajectory lengths (shorter lengths as you said). But it might not be appropriate to claim this as the contribution of your work.

---

> > > > > ### Author Response · Authors · 2018-10-29
> > > > > **Difference in beam size and their performance**
> > > > >
> > > > > Hi,
> > > > >
> > > > > Thanks for the opportunity to further clarify our usage of a smaller beam size (15 as opposed to 40).
> > > > >
> > > > > We do not claim that using a smaller number of beams is one of the major contributions. It was a nice side effect that resulted from our progress monitor, where we can evaluate the partial and unfinished candidate routes during beam search. As a result, we are able to maintain a lower number of beams but still achieve state-of-the-art success rate.
> > > > >
> > > > > Below are the results with different beam size for reference.
> > > > >
> > > > >
> > > > > --------------------------------------------------------------------------------------------------------------------------------------------------------------------------------------------------------
> > > > > 			Co-Grounding	  Progress    Data			        Beam 	  		Validation-Seen 	                        Validation-Unseen
> > > > > 		    #  Visual	Textual     Monitor     Augmentation	Search (size) 	length↓   NE↓    SR↑    OSR↑ 	length↓     NE↓     SR↑     OSR↑
> > > > > --------------------------------------------------------------------------------------------------------------------------------------------------------------------------------------------------------
> > > > > 		    1         ✔  	     ✔		✔ 		  ✔ 	  		        5		  		159.19	3.03    0.71    0.79 	        168.13	   4.77     0.55     0.68
> > > > > Ours	    2         ✔  	     ✔		✔ 		  ✔ 	  		        10		  		271.51	3.11    0.71    0.78 	        277.13	   4.64     0.57     0.68
> > > > > 		    3         ✔  	     ✔		✔ 		  ✔ 	  		        15		  		355.13	3.04    0.71    0.78 	        360.46	   4.62     0.58     0.68
> > > > > --------------------------------------------------------------------------------------------------------------------------------------------------------------------------------------------------------

---

> > > > ### Public Comment · (anonymous) · 2018-11-01
> > > > **Contradictions of statistics reported above**
> > > >
> > > > The 5th row reported above clearly corresponds to 2nd row of Table 2 in your paper:
> > > > 1  3.65 0.65 0.75 6.07 0.42 0.57
> > > > 2  3.23 0.70 0.78 5.04 0.57 0.70
> > > > 3  3.04 0.71 0.78 4.62 0.58 0.68
> > > >
> > > > So all the reported numbers in Table 2 use beam search. However, 1st row in Table 2 matches entirely with 3rd row reported above (has no beam search),  which is contradicting and confusing.
> > > >
> > > > In the table below, you claimed the proposed approach is better than Speaker-Follower with or without beam search. However, the following comparison was also misleading, as you used beam size 15. Why can't you adopt the same beam size and compare? On the other hand, the 61% SR is 4% boost compared to no data-augmentation counterpart in Table 1 of paper, while on validation-unseen, the gap is merely 1%, any idea why?
> > > >
> > > > 									with beam search
> > > > --------------------------------------------------------------------------------------------------------------------------------------
> > > > Speaker-Follower		   	         1257.38    4.87     0.53    0.96   	0.01
> > > > Ours			   			 373.09      4.48     0.61    0.97   	0.02

---

> > > > > ### Author Response · Authors · 2018-11-02
> > > > > **Only progress monitor use beam search during inference**
> > > > >
> > > > > Hi,
> > > > >
> > > > > We are sorry if there is any confusion regarding table 2 in the paper. As stated in the Sec. 2.3 of “Progress Monitor” in the paper, during inference we use beam search with the progress monitor. Therefore, the 1st row in our proposed method with only co-grounding does not use beam search, which outperformed baseline with panoramic action space by 15% on validation-unseen SR. We hope that the ablation study table above clarifies this.
> > > > >
> > > > > In the comment below, we have shown that, with beam size 5, our proposed method already achieved the state-of-the-art performance. By further increasing beam size, the performance gradually increases until it saturates.
> > > > >
> > > > > With panoramic action space, many of the beams are actually empty when using a very large beam size due to limited navigable directions per viewpoints. Thus, increasing the beam size to a larger number will not necessarily help if the competing between beams already provides good selections (we achieved this by leveraging the progress monitor). Nonetheless, we provide the result with a beam size 40 as per requested.
> > > > >
> > > > > ----------------------------------------------------------------------------------------------------------------------------------------------------------------------
> > > > > 			Co-Grounding	  Progress    Data			        Beam 	  		Validation-Seen 	    Validation-Unseen
> > > > > 		    #  Visual	Textual     Monitor     Augmentation	Search (size) 	NE↓    SR↑    OSR↑      NE↓     SR↑     OSR↑
> > > > > ----------------------------------------------------------------------------------------------------------------------------------------------------------------------
> > > > > Ours		   ✔  	     ✔		✔ 		  ✔ 	  		        40		  		3.13    0.70    0.77 	    4.51    0.58     0.68
> > > > > ----------------------------------------------------------------------------------------------------------------------------------------------------------------------

---

> > > > ### Public Comment · (anonymous) · 2018-11-08
> > > > **Result is not clear**
> > > >
> > > > 1. It seems that the performance with "Textual only" is 1% beyond the Co-Grounding model. So why not use the "textual only" in the further experiments?
> > > >
> > > > 2. Could you give more details on the method that you submitted to the test server for the "Test-Unseen" result? Is it "Co-Grouding" or "Co-Grounding + Progress Monitor"?
> > > > If the progress monitor is included, is it still non-beam search result? Because the progress monitor is only used in beam-search according to your paper.
> > > >
> > > > Please let me know if I have any misunderstanding.

---

> > > > > ### Author Response · Authors · 2018-11-22
> > > > > **Please see the updated ablation study table in our revision**
> > > > >
> > > > > Hi,
> > > > >
> > > > > 1.
> > > > > Although removing visual grounding may seem to produce a slightly higher performance on unseen SR, We chose to use both visual and textual grounding since the training is more stable and the model is less prone to overfitting.
> > > > >
> > > > > 2.
> > > > > The test server result for "without beam search" is using "co-grounding + progress monitor". Please kindly see our updated ablation study table in our revision for further details.

---

### Public Comment · (anonymous) · 2018-10-24
**Panoramic Action Space**

Hi, thanks for the good performance!

I am wondering how much the panoramic action space helps in your model. Can you report the performance without the panoramic action space? Thanks!

Best regards

---

> ### Author Response · Authors · 2018-10-25
> **We use the model with panoramic action space as baseline**
>
> Hi,
>
> Thank you for your interest in our work and kind words!
>
> Our proposed method built upon the established work, and we use the panoramic action space proposed in the Speaker-Follower as the baseline (hence making the comparison to our improvements fair). We focus on highlighting the novel contributions/ideas that improve on this baseline. Removing the panoramic action space from the proposed method requires non-trivial changes to be made, including visual grounding module, action selection module, and finally the progress monitor itself. We thus encourage readers to refer to the Speaker-Follower paper for performance improvement regarding panoramic action space.

---

### Public Comment · (anonymous) · 2018-10-24
**Confusions about Eq 6**

Hi interesting paper! I am trying to reproduce the result and is having problem with Eq 6.
Specifically, the progress monitor module seems to output p_t^pm which seems to be a 1D value between -1~1 after tanh(). The target y_t^pm is also a 1D number that is between -inf ~ 1.
In this case how do you use CrossEntropy Loss as suggested in the paper?
Eq 6 suggested that the loss should incorporate - y_t^pm * log (p_t^pm). Well I am confused if this is still "cross entropy loss". What's more log() does not like negative values?

---

> ### Author Response · Authors · 2018-10-25
> **Thank you for bringing this to our attention**
>
> Hi,
>
> Thank you for your interest in our paper and trying to reproduce it!
>
> You are correct. The cross entropy loss in Eq 6 should be a Mean Squared Error loss (MSELoss). The equation should thus be changed to \sum_{t=1}^{T} (y^{pm}_{t} - p^{pm}_{t})^2.  The original cross entropy loss was a variant we experimented with predicting whether the agent is making progress or not (binary prediction). The current version with tanh() at the output of the progress monitor and trained with MSE loss gave us better performance.
>
> We will correct this error in the revision.

---

> > ### Public Comment · (anonymous) · 2018-10-26
> > **What about the variable length text attention weights?**
> >
> > Thanks so much for the prompt reply.
> >
> > One more question regarding Eq6: how do you handle the text attention weights, alpha? The supplementary material suggests that W_pm is one linear layer of shape 592 x 1. The text attention weights, however, is produced from various lengths of instructions and can have variable lengths. It seems a bit odd to simply pad zeros at the end of it to extend it to the fixed length 80. How did you handle it?

---

> > > ### Author Response · Authors · 2018-10-30
> > > **Zero-padding for various lengths of instructions**
> > >
> > > Hi,
> > >
> > > Thanks for the comment.
> > >
> > > Zero-padding is exactly how we handled it for various lengths of instructions. We have also explored ideas like using interpolation to upsample the attention weights of short instructions to a fixed length of 80, but it produces a similar performance as without interpolation.

---

> > > > ### Public Comment · (anonymous) · 2018-11-05
> > > > **Training Progress Monitor**
> > > >
> > > > Hi,
> > > >
> > > > Thanks a lot for your responses. My reproduction of co ground model (without progress monitor) achieved comparable performance to reported. However, training the progress monitor seems pretty hard. Mind answering a few more questions?
> > > > - Did you include data augmentation in the reported 46% for co-ground + progress monitor?
> > > > - Any chance you could verify EQ5?
> > > > - It seems from EQ6 that you used a training loss equal to the sum of loss on any trajectory. Did you consider using per-step loss? Did you take the average loss across batch?
> > > > - The scale of co-ground loss and progress-monitor loss are roughly 10:1. Is this expected for training?

---

> > > > > ### Author Response · Authors · 2018-11-22
> > > > > **Thanks for reproducing our results**
> > > > >
> > > > > Hi,
> > > > >
> > > > > Thanks for reproducing our results using the co-grounding module.
> > > > >
> > > > > 1.
> > > > > The 46% reported in the comment is without data augmentation. Originally, we only use data augmentation for test server submission, and all ablation study settings are without data augmentation unless specified. Please see the updated ablation study table in our revision for further details on the performance reported.
> > > > >
> > > > > 2. The Eq. 5 seems to be correct. Can you please elaborate more if you have any concerns?
> > > > >
> > > > > 3. The losses across time step are summed together. Please note that the loss should be zero for the samples which are ended. Also, if the distance to the goal is lower than 3, we set the target y^{pm}_t to 1 (we have clarified this in the revision).
> > > > >
> > > > > 4. Yes, it is roughly 10:1 during training.

---

### Meta-Review · Area_Chair1 · 2018-12-15

**Recommendation:** Accept (Poster)
**Confidence:** 4

**Metareview:**

The authors have described a navigation method that uses co-grounding between language and vision as well as an explicit self-assessment of progress. The method is used for room 2 room navigation and is tested in unseen environments. On the positive side, the approach is well-analyzed, with multiple ablations and baseline comparisons. The method is interesting and could be a good starting point for a more ambitious grounded language-vision agent. The approach seems to work well and achieves a high score using the metric of successful goal acquisition. On the negative side, the method relies on beam search, which is certainly unrealistic for real-world navigation, the evaluation metric is very simple and may be misleading, and the architecture is quite complex, may not scale or survive the test of time, and has little relevance for the greater ML community. There was a long discussion between the authors and the reviewers and other members of the public that resolved many of these points, with the authors being extremely responsive in giving additional results and details, and the reviewers' conclusion is that the paper should be accepted.